# The non-template functions of helper virus RNAs create optimal replication conditions to enhance the proliferation of satellite RNAs

**Zimu Qiao, Jin Wang, Kaiyun Huang, Honghao Hu, Zhouhang Gu, Qiansheng Liao, Zhiyou Du** \*

College of Life Sciences and Medicine, Zhejiang Sci-Tech University, Hangzhou, Zhejiang, China

\* duzy@zstu.edu.cn

**Data Availability Statement:** All relevant data are within the manuscript and its Supporting Information files.

## Abstract

As a type of parasitic agent, satellite RNAs (satRNAs) rely on cognate helper viruses to achieve their replication and transmission. During the infection of satRNAs, helper virus RNAs serve as templates for synthesizing viral proteins, including the replication proteins essential for satRNA replication. However, the role of non-template functions of helper virus RNAs in satRNA replication remains unexploited. Here we employed the well-studied model that is composed of cucumber mosaic virus (CMV) and its associated satRNA. In the experiments employing the CMV *trans*-replication system, we observed an unexpected phenomenon the replication proteins of the mild strain LS-CMV exhibited defective in supporting satRNA replication, unlike those of the severe strain Fny-CMV. Independent of translation products, all CMV genomic RNAs could enhance satRNA replication, when combined with the replication proteins of CMV. This enhancement is contingent upon the recruitment and complete replication of helper virus RNAs. Using the method developed for analyzing the satRNA recruitment, we observed a markedly distinct ability of the replication proteins from both CMV strains to recruit the positive-sense satRNA-harboring RNA3 mutant for replication. This is in agreement with the differential ability of both 1a proteins in binding satRNAs in plants. The discrepancies provide a convincing explanation for the variation of the replication proteins of both CMV strains in replicating satRNAs. Taken together, our work provides compelling evidence that the non-template functions of helper virus RNAs create an optimal replication environment to enhance satRNA proliferation.

## Author summary

Satellite RNAs (satRNAs) are a type of subviral pathogens that encode neither replication protein nor coat protein. Consequently, they must compete with helper virus RNAs for viral proteins and host resources to ensure their survival. The competition relationship has led us to conceptualize that helper virus RNAs might exhibit antagonism with satRNAs during replication. Several studies have highlighted the importance of non-template functions of viral RNAs in viral replication, such as viral replicase-mediated RNA

**Funding:** This work was supported by the grants from the National Natural Science Foundation of China with the numbers (31870144, 32070154) to Z.D. The funder had no role in study design, data collection and analysis, decision to publish, or preparation of the manuscript.

**Competing interests:** The authors have declared that no competing interests exist.

recruitment and participating in the assembly of viral replication complex. However, it remains unknown whether this significance extends to satRNA replication. To address this question, we utilized a well-established model involving cucumber mosaic virus and its associated satRNAs. Our data provides compelling evidence to uncover the biological relevance of the non-template functions of helper virus RNAs in enhancing satRNA replication. Thus, our work prompts a reconsideration of the molecular interplay between helper virus RNAs and satRNAs in their replication, previously thought to be mutually competitive.

## Introduction

Satellite RNAs (satRNAs) are among the smallest infectious agents, encoding neither RNA polymerase nor coat protein (CP), thus relying entirely on their helper virus to complete their infection cycles. In most cases, satRNAs are non-coding RNA molecules with limited or no sequence similarity to their cognate helper virus [1]. These features render them ideal models for investigating the structure, function, and evolution of pathogenic RNAs, as well as their interactions with helper viruses and host organisms [1–4]. A particularly appealing aspect for researchers is the satRNA capability to ameliorate disease symptoms caused by their helper viruses, underscoring their potential as tools for antiviral bio-control [5–7]. Despite substantial achievements that have been made already [2–4,8–14], some pivotal questions surrounding satRNAs remain unanswered, such as biologically supported RNA structure and its interplay with viral proteins, which impedes a comprehensive understanding of satRNA functions and potential applications.

Except for a limited number of mycovirus-associated satRNAs [15–17], the majority of satRNAs are associated with plant-infecting positive sense (+) RNA viruses [14], which represent the largest category of plant viruses and pose significant threats to global agriculture. Upon infection, these (+) RNA viruses release their genomic RNAs to produce viral replication proteins. These replication proteins then localize to specific subcellular membranes, where they recruit viral RNAs along with numerous host factors to remodel the co-opted endomembrane, ultimately giving rise to structures known as viral replication organelles (VROs) [18–23]. The recruitment of viral RNAs into these VROs can occur through two distinct mechanisms. The first mechanism involves the recognition of replication proteins to specific RNA elements referred to as recruitment recognition elements (RREs) [24–29]. The second mechanism is RRE-independent and is coupled with translation [25,30]. Viral RNAs, in addition to serving as templates for translation, also serve as scaffolds, facilitating the assembly of functional viral replication complexes (VRCs) through molecular interactions with viral replication proteins and host proteins [31–33]. Thus, these non-template functions of viral RNAs, involved in viral RNA recruitment and VRC assembly, play important roles in viral RNA replication. Regarding satRNA replication, genomic RNAs from their cognate helper virus function as translating templates, providing the necessary replication proteins. However, the roles of the non-template functions of helper virus RNAs in satRNA replication remain unexploited.

Cucumber mosaic virus (CMV) holds a pivotal place in the world of plant pathogens due to its substantial economic and scientific significance [34–36]. CMV possesses a tripartite (+)-sense RNA genome, encompassing RNAs 1–3, which are terminated with a tRNA-like structure (TLS) at their 3′ termini. The 1a protein, a product of RNA1, localizes to the tonoplast [37,38], where it orchestrates the recruitment of the RNA2-encoded 2a protein (RNA-dependent RNA polymerase, RdRP) and host factors, along with viral genomic RNAs,

culminating in the formation of VRCs [39,40]. RNA2 harbors another functional protein, 2b, situated proximally to its 3′ end, which plays the role of an RNA silencing suppressor and stands as a critical virulence determinant [41–46]. RNA3 encodes two separate open reading frames (ORFs): movement protein (MP) and CP. Both are indispensable for facilitating virus mobility within plants, with the CP additionally responsible for packaging viral RNAs and, if present, satRNAs [34].

To date, numerous CMV strains have been reported and categorized into subgroups I and II, exhibiting significant variations in RNA sequence [47]. Many CMV strains have been found to associate with satRNAs, which are linear, single-stranded RNA molecules spanning a range of 330 to 405 nucleotides (nt) [3,34]. CMV satRNAs have served as a valuable model for elucidating the mechanisms underlying the replication of pathogenic RNAs. The replication of satRNAs hinges on the replication proteins produced by RNA1 and RNA2 of CMV [48]. Purified viral replicases derived from CMV-infected leaves have demonstrated their capability to support the complete replication of satRNAs *in vitro* [49,50]. This is consistent with the findings from the *in trans* replication assays, which reveal that CMV replication proteins, generated from non-replicable RNA transcripts, can effectively support the replication of satRNA isolate T1 (sat-T1) [51]. These studies collectively indicate that the non-template functions of viral RNAs are dispensable for satRNA replication. In addition, *in vitro* replication assays reported that viral RNAs compete with satRNAs for limited amounts of purified viral replicases, resulting in decreased replication levels of viral and satellite RNAs [52]. In fact, CMV satRNAs usurp viral replicases effectively for their replication, leading to a high accumulation level. This hints that satRNAs possess potential mechanisms to prompt their replication.

In this study, we explored the role of the non-template functions of helper virus RNAs in satRNA replication, using the well-established model featuring CMV and its associated satRNA. In *trans*-replication experiments with satRNAs, we made an intriguing finding that the replication proteins of the mild strain LS-CMV (subgroup II) exhibited defective in supporting satRNA replication, unlike those of the severe strain Fny-CMV (subgroup I). Viral genomic RNAs lacking the expression of viral proteins had the capability to enhance satRNA replication when combined with the replication proteins of Fny-CMV or LS-CMV. This enhancement was determined to be dependent on the recruitment and complete replication of viral RNAs. Interestingly, we found that LS replication proteins possessed a limited ability to recruit (+) strands of satRNAs. Moreover, LS 1a protein was not proficient as Fny 1a protein in interaction with satRNAs. These findings provide a convincing explanation for its diminished ability to replicate satRNAs in the absence of viral RNAs. Collectively, our work provides compelling firsthand evidence that underscores the importance of the non-template functions of helper virus RNAs in promoting satRNA replication. Our work will help us to re-think over the relationship between helper virus RNAs and satRNAs during their replication, previously suggested to be mutually competitive [52].

## Materials and methods

### Infectious constructs of CMV, satRNAs and their derivatives

The T-DNA-based infectious plasmids pCB301-C1, pCB301-C2 and pCB301-C3 for genomic RNAs 1–3 of Fny-CMV, and pCB301-LS109, pCB301-LS209, and pCB301-LS309 for LS-CMV, have been documented previously [53,54]. The plasmids pCB301-C1a and pCB301-C2a, used for transient expression of the 1a and 2a proteins of Fny-CMV, were also generated in the previous work [53]. To transiently express the 1a or 2a protein of LS-CMV, pCB301-L1a and pCB301-L2a were constructed by amplifying the coding sequences of LS 1a

and 2a, and separately inserting them into pCB301 after digestion with *Stu*I and *Sac*I, as reported previously [53]. All the mutants, featuring mutations or deletions in these infectious clones, were generated using one-step site-directed mutagenesis reported previously [55]. To prevent expression of the 2b protein from LS RNA2, we substituted both $_{2423}$TTG$_{2425}$ and $_{2438}$TCG$_{2440}$ with TAG in pCB301-LS209, to generate pCB301-L2Δ2b. It is worth mentioning that the numbers flanking the codons mentioned here and below correspond to their locations in viral genomic RNAs. The constructed plasmids, pCB301-ncL1, pCB301-ncL2, and pCB301-ncL3, represent non-coding variants corresponding to RNA1, RNA2, and RNA3 of LS-CMV, respectively. To generate pCB301-ncL1, both codons $_{415}$ATG$_{417}$ and $_{442}$ATG$_{444}$ were substituted with TGA in pCB301-LS109. pCB301-ncL2 was created by replacing the initiation codon of 2a with AAG, $_{471}$ATG$_{472}$ with TGA, and $_{485}$TTG$_{487}$ with TAG in pCB301-L2Δ2b. pCB301-ncL3 was produced by inserting five nucleotides (TAGTA) and four nucleotides (TGAT) immediately downstream of the initiation codon of the MP and the CP ORFs in pCB301-LS309, respectively. The same insertions were introduced into pCB301-C3, resulting in pCB301-ncF3. The noncoding versions of RNA1 (RNA1Δ1a) and RNA2 (RNA2Δ2a) of Fny-CMV were reported previously [51], renamed as ncF1 and ncF2 in this work, respectively. Similarly, the non-coding versions (ncT1, ncT2, ncT3) corresponding to genomic RNAs 1–3 of tomato aspermy virus (TAV) were generated by introducing the mutations as shown in the Result section, into the infectious clones reported previously [53]. To generate the plasmids pCB301-L4 and its non-coding version pCB301-ncL4, the sequence of LS RNA3 spanning from 1 to 1167 nt or from 1 to 1172 nt was deleted in the plasmid pCB301-LS309 and pCB301-ncL3, respectively. Additional constructs with mutations or deletions in pCB301-LS309 are presented schematically in the Result section.

The ligation-independent cloning (LIC) method reported previously [56] was employed to insert a DNA fragment in a designated plasmid to create chimeric or recombinant viral RNAs. For instance, commercially synthesized cDNA fragments corresponding to the TLS of tobacco mosaic virus (TMV), tobacco yellow mosaic virus (TYMV), brome mosaic virus (BMV), TAV, or peanut stunt virus (PSV) were used to replace the TLS in pCB301-LS309, to generate five chimera: pCB301-ncL3-TLS$^{TMV}$, pCB301-ncL3-TLS$^{TYMV}$, pCB301-ncL3-TLS$^{BMV}$, pCB301-ncL3-TLS$^{TAV}$, or pCB301-ncL3-TLS$^{PSV}$, respectively. The cDNA fragments corresponding to the (+)-sense or (-)-sense RNA of sat-T1 were amplified through regular polymerase chain reactions (PCRs) and inserted downstream of the CP ORF in pCB301-mF3 using the LIC method, giving rise to pCB301-mF3-T1(+) and pCB301-mF3-T1(-), respectively. pCB301-mF3 is the derivative of pCB301-C3, in which the Box-B motif was substituted with the nucleotides 5′TTCCATTCCAA3′. Employing the same approach, a 337-nt DNA fragment of the β-glucuronidase (GUS) gene was inserted into pCB301-F3 or pCB301-mF3, resulting in pCB301-F3-gus and pCB301-mF3-gus, respectively. The creations of the infectious DNA constructs pCB301-sat-T1, pCB301-sat-D4 and pCB301-sat-SD have been documented in our previous work [51]. Similarly, the infectious constructs pCB301-sat-RS, pCB301-sat-Y, pCB301-sat-Yi, and pCB301-sat-KN were generated by inserting each of the commercially synthesized cDNA fragments of these satRNAs into pCB301 pre-digested by restriction endonucleases *Stu*I and *Sma*I. The nucleotide sequences of these satellite strains RS, Y, Yi and KN are available in GenBank with their IDs: AF451896, D00542, DQ412733, and D28559, respectively.

To transiently produce the RNA transcripts 6×MS2-satT1 or 6×MS2-gus in plants, the DNA fragment corresponding to the 6 copies of MS2 CP (MCP) binding stem-loop as reported previously [57], was synthesized commercially and inserted into pBI121 after digestion with *Xba*I and *Bam*HI. Subsequently, the cDNA fragment of sat-T1 or a 337-nt DNA fragment of the GUS gene was introduced downstream of the 6×MS2 sequence after digestion

with *BamH*I and *Sac*I, to generate pBI121-6×MS2-satT1 or pB1121-6×MS2-gus, respectively. To express MCP fused with yellow fluorescence protein (YFP) and a nuclear localization signal (NLS), YFP with the second NLS of LS-CMV 2b was amplified using regular PCR reactions, and inserted into pCambia-1300 after digestion with restriction enzymes *Nco*I and *Sac*I. Subsequently, the DNA fragment corresponding to the modified MCP containing two mutations (V75Q and A81G) as reported previously [58], was synthesized commercially and introduced upstream of YFP after digestion with restriction enzymes *BamH*I and *Nco*I. To express F1a or L1a tagged with mCherry, the coding sequences of F1a and L1a without a stop codon were amplified using regular PCR reactions and separately inserted into pBI121 between *BamH*I and *Sac*I. In this step, a *Kpn*I cloning site was introduced at the upstream of *Sac*I. Then, the DNA fragment of mCherry was inserted downstream of F1a or L1a after digestion using restriction enzymes *Kpn*I and *Sac*I, generating pBI21-F1a-mCherry and pBI121-L1a-mCherry, respectively.

Prior to transformation into *Agrobacterium tumefaciens* GV3101, all plasmids underwent sequencing for authentication. All the primers used for constructing these plasmids in this study are listed in the Supplemental data (S1 Table).

### *Agrobacterium*-mediated virus inoculation in *N. benthamiana*

*N. benthamiana* plants were cultivated in a plant growth room that maintained a 16-hour photoperiod and provided light with an intensity of 150 to 200 mE/m$^2$/s at 22–24°C. For virus inoculation, we employed agroinfiltration on approximately four-week-old plants, following the procedure described previously [51]. Briefly, *Agrobacterium* cells harboring an infectious clone of LS-CMV or satRNAs were first propagated in Luria-Bertani (LB) liquid media supplemented with 50 ng/ml kanamycin and 25 ng/ml rifampin. After propagation, bacterial cells were harvested and adjusted to an optical density of 1.0 at a wavelength of 600 nm in infiltration solution composed of 10 mM MgCl$_2$, 10 mM 2-(*N*-morpholino) ethanesulphonic acid, and 100 mM acetosyringone. The bacterial cells carrying pCB301-LS109, pCB301-LS209 or pCB301-LS309, as well as the cells carrying the vector (pCB301) or the infectious plasmid of satRNAs, were mixed with an equal proportion, and incubated in darkness at room temperature for 3 hours. Subsequently, the mixed cells were infiltrated into the 5$^{th}$ leaves of *N. benthamiana* plants. In parallel, the plants treated with infiltration solution served as the mock control. At 3 days post-agroinfiltration (DPAI), the infiltrated leaves from three plants were collected for the analyses of the accumulation levels of viral RNAs and satRNAs by northern blot hybridization as described below.

### *Trans*-replication assays in *N. benthamiana* plants

The *trans*-replication system, a method previously described for studying CMV replication in *N. benthamiana* plants [51,59], was employed in this study to assess the influence of viral RNAs and their variants on satRNA replication. Briefly, the *Agrobacterium* cells harboring the plasmids used for transiently expressing the RNA silencing suppressor P19, CMV replication proteins, and one of satRNAs, were mixed with an equal proportion, along with the bacterial cells carrying either an empty vector (pCB301) or a plasmid expressing one of viral RNAs. The mixed bacterial cells were incubated in dark at room temperature for 3–4 hours and subsequently infiltrated into the 6$^{th}$ true leaves of approximately four-week old *N. benthamiana* plants. At 3 DPAI, the infiltrated leaves were collected for extraction of total RNAs, followed by northern blotting analyses of the accumulation levels of viral and satellite RNAs as described below.

## Confocal microscopy

The 6[th] true leaves of *N. benthamiana* plants with approximately 4-weeks old were infiltrated with *Agrobacterium* cells to co-express MCP-YFPnls and either F1a-mCherry or L1a-mCherry, along with 6×MS2-satT1 or 6×MS2-gus. At 2 DAPI, the infiltrated leaves were collected for visualization of fluorescence using a laser confocal microscopy (Zeiss, LSM880), and the confocal images were processed with the software ZEN (Zeiss).

## Northern blot hybridization

Northern blot hybridization was carried out for analyses of viral and satellite RNAs according to the procedure described previously [51]. Briefly, total RNAs were extracted from the infiltrated leaf tissues using RNA extraction buffer (0.05 M NaOAc pH 5.2, 0.01 M EDTA pH 8.0, and 1% SDS) as described [51]. Two µg of total RNAs were separated on a 1.5% agarose gel containing 7% formaldehyde. The separated RNAs were transferred onto a positively changed nylon membrane (GE) and hybridized with the digoxin (DIG)-labeled DNA oligonucleotide probes, specifically designed for detecting CMV genomic RNAs or satRNAs [51,59]. In addition, the oligonucleotide probe used for detecting the LS RNA3 mutants lacking the conserved sequence in the 3′ untranslated region (3′ UTR) is complementary to the sequence positioning from 1299 to 1333 nt of LS RNA3. The oligonucleotide probe (5′TAGACATTCACGGAGAT CAGCATAGC3′), denoted as probe-wtsat, employed for the simultaneous detection of seven distinct satRNAs. Other probes used in specific experiments were mentioned elsewhere. DIG-labeled DNA probes were detected by a chemiluminescence-based DIG detection kit (Roche) following the manufacturer's instructions.

## Strand-specific reverse transcription-PCR (RT-PCR)

Strand-specific RT-PCR was employed to detect (-)-sense RNAs of LS RNA3 and its mutants using the procedures reported previously [39], with slight modifications. Briefly, the primer ncL3a-F, complementary to the (-)-sense RNA of LS RNA3, was used to prime synthesis of the first strand cDNAs in the presence of the reverse transcriptase Superscript III (Invitrogen). The generated cDNAs were subjected to PCR reactions with the primer pair ncL3a-F/ncL3b-R using Q5 high-fidelity DNA polymerase (NEB). The PCR reactions were conducted with the process, which was composed of an initial denaturation step at 98°C for 3 min, followed by 30 cycles of 10 sec at 98°C, 15 sec at 58°C, and 30 sec at 72°C, with a final extension step for 5 min at 72°C. Finally, PCR products were separated in a 1% agarose gel, and visualized under UV light after staining with ethidium bromide. Both primers ncL3a-F and ncL3b-R are listed in the supplemental data (S1 Table).

## Results

### Unlike the replication proteins of Fny-CMV, those of LS-CMV are defective in supporting satRNA replication in *trans*-replication assays

Previously, we reported the successful infection of sat-T1 in the presence of the severe strain Fny-CMV (subgroup I) [51]. Here we examined the accumulation of sat-T1 in the presence of the mild strain LS-CMV (subgroup II), as well as Fny-CMV, in the infiltrated leaves of *N. benthamiana* plants at 3 DPAI. Northern blotting analyses showed that both polarities of sat-T1 were undetectable in the absence of CMV (vector), even upon over-exposure for both strands (Fig 1A). This suggests that the RNA transcripts of sat-T1 and potentially amplified products by host RNA polymerases, had low accumulation levels, falling below the detection limit of the method we employed. In the presence of Fny-CMV or LS-CMV, each polarity of

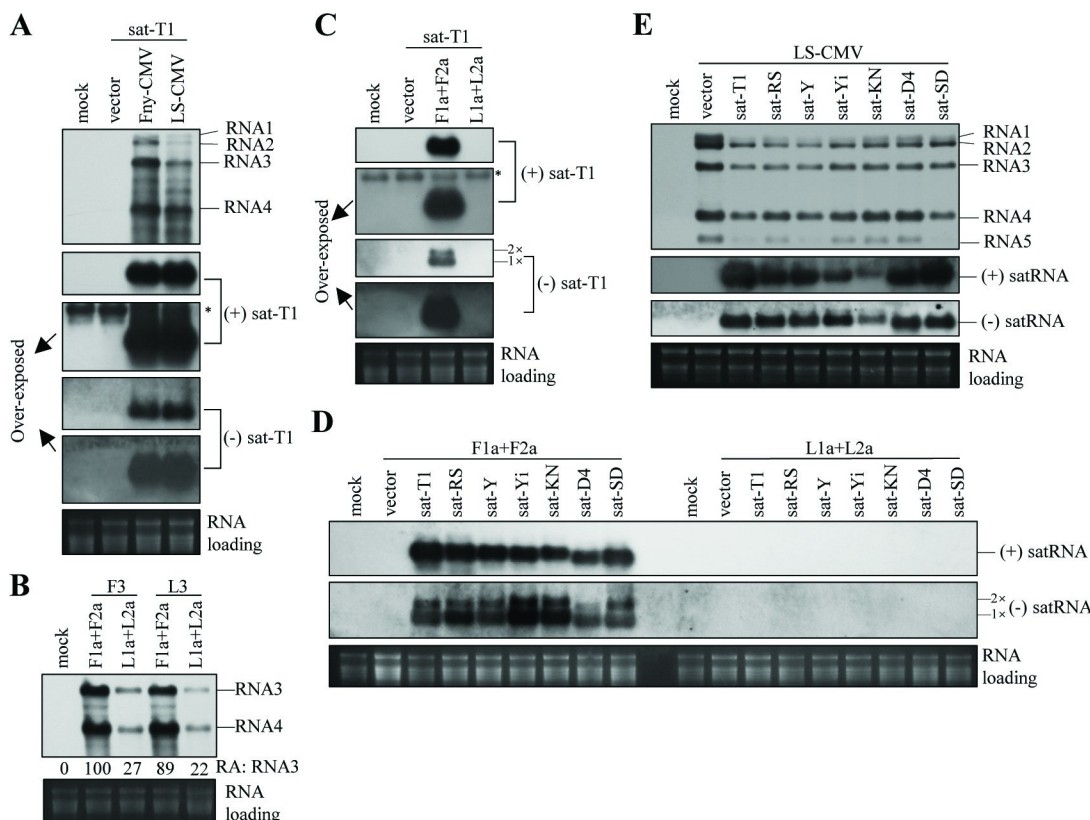

**Fig 1. Replication proteins from Fny-CMV, but not LS-CMV, efficiently support the replication of satRNAs in *trans*-replication assays.** (A) Northern blotting analyses of the accumulation of CMV RNAs and sat-T1 in the infiltrated leaves. (B) Northern blotting analyses of the accumulation of RNA3 and its subgenomic RNA4. In the *trans*-replication assays, RNA3 from Fny-CMV (F3) or LS-CMV (L3) was replicated by the replicase of both CMV strains (F1a and F2a, L1a and L2a), in combination with the RNA silencing suppressor P19 from tomato bushy stunt virus. (C, D) Northern blotting analyses of the accumulation of sat-T1 (C) and seven wild-type satRNA strains (D). Sat-T1 and the other six satRNAs were replicated with the replication proteins of Fny-CMV or LS-CMV, along with P19. (E) Northern blotting analyses of viral RNAs and satRNAs in the co-infected plants. Seven satRNA strains were separately co-inoculated with LS-CMV on *Nicotiana benthamiana* plants via agroinfiltration. Both the *trans*-replication assays (A-D) and virus inoculation (E) were conducted by infiltrating the 5th true leaves of *N. benthamiana* plants with *Agrobacterium* cells. Total RNAs were extracted from the infiltrated leaves at 3 days post-agroinfiltration (DPAI) (A-D) or from the upper systemic leaves at 6 DPAI (E), followed by northern blot hybridization using the CMV- or satRNA-specific oligonucleotide probe. Mock plants were treated with the infiltration solution. Vector controls in (A, C-E) refer as to these samples infiltrated with *Agrobacterium* cells carrying the vector pCB301, in combination with sat-T1 (A and C), viral replication proteins (D) or LS-CMV (E). Asterisk (*) denotes the non-specific bands shown after over-exposure in (A) and (C). Ethidium bromide-stained ribosomal RNAs were used to assess the relative loading amounts of each RNA sample.

sat-T1 was readily detected with nearly equal levels between both CMV strains, although LS-CMV RNAs accumulated to a lesser extent compared to Fny-CMV (Fig 1A).

Our previous work has demonstrated that the replication proteins, specifically 1a and 2a, of Fny-CMV, effectively support sat-T1 replication without the assistant of replicable viral RNAs in a *trans*-replication system [51]. Here we tested the replication proteins of LS-CMV and Fny-CMV in promoting the replication of CMV RNA3 or sat-T1 in the infiltrated leaves of *N. benthamiana* plants at 3 DPAI, as described previously [51]. The replicase of Fny-CMV, as well as LS-CMV, replicated both homologous and heterologous RNA3s with similar efficiency. Nevertheless, the replicase of Fny-CMV produced notably higher quantities of RNA3 and RNA4, compared to its LS counterpart (Fig 1B). The difference in the accumulation levels of these viral RNAs produced by both viral replicases would be due to the distinct replication

efficiency, as the RNA silencing suppressor p19 was included to suppress antiviral RNA silencing in this assay. As previously reported [51], the replicase of Fny-CMV efficiently multiplied sat-T1, evidenced by the considerable accumulation of (+) strands of sat-T1, as well as the two bands specific to (-) strands of sat-T1, which were presumed to be monomer and dimer, respectively (Fig 1C). Strikingly, sat-T1 was undetectable in the leaves transiently expressing LS replication proteins, similar to the vector control, even upon over-exposure (Fig 1C). We then wondered whether LS replication proteins were also defective in replicating other satRNA strains. Seven satRNA strains, exhibiting extensive variations in sequence similarity (ranging from 76.9%-98.8%) and genome size (ranging from 337 nt-405 nt) were tested in combination with the replication proteins of Fny-CMV or LS-CMV. Similar to sat-T1, the other six strains were efficiently proliferated in the presence of Fny replication proteins, showing one specific band of (+)-stranded RNAs for each satRNA strain, as well as the two bands for (-)-stranded RNAs of these satRNAs (Fig 1D). In contrast, none of the seven satRNA strains were proliferated by LS replication proteins at a detectable level, nor in the vector control (Fig 1C). These findings suggest that unlike the replication proteins of Fny-CMV, those of LS-CMV require additional viral components to support satRNA replication. Indeed, when co-inoculated with LS-CMV in *N. benthamiana* plants, all seven satRNAs were successfully detected in the upper systemic leaves with substantial accumulations of both polarities of these satRNAs (Fig 1E). Taken together, our results indicate a clear variation in the capability of the replication proteins of Fny and LS to support satRNA replication when viral RNAs are absent. Furthermore, in addition to the replication proteins, other viral components are necessary for LS-CMV to stimulate satRNA replication.

## Replicable viral RNAs play vital roles in enhancing satRNA replication, independent of translation products

The successful replication of these seven satRNAs in the presence of LS-CMV prompted us to examine which viral RNA(s) was necessary for LS replication proteins to support efficient replication of satRNAs. In the *trans*-replication system, where LS replication proteins were expressed, sat-T1 was co-expressed with RNA1 (L1), RNA2Δ2b (L2Δ2b, lacking 2b expression), or RNA3 (L3) of LS-CMV. RNA gel blotting results showed that each viral RNA was effectively replicated and enabled viral replicases to amplify both polarities of sat-T1, albeit with varying efficiency, particularly in the plus polarity (Fig 2B). Given the distinct protein products translated from L1, L2Δ2b, and L3, we hypothesized that the facilitation of sat-T1 by these RNAs was due to the RNAs themselves, rather than their translation products. The hypothesis was confirmed by evidence that these noncoding variants (ncL1, ncL2, ncL3) of LS RNAs 1–3 (Fig 2A) exhibited nearly equal efficiency in stimulating sat-T1 replication in the *trans*-replication assays (Fig 2C). Thus, among the coding versions of viral RNAs (Fig 2B), the least capability of L2Δ2b to stimulate sat-T1 replication could be attributed to the over-production of the 2a protein translated from the replicable RNA2 mutant, potentially disrupting the balance between the 1a and 2a proteins during replication. Comparing the efficiencies of L3 and ncL3 in stimulating sat-T1 replication (Fig 2B & 2C), it appears that L3 was more efficient than ncL3. In some (+)-strand RNA viruses, CP has been well-documented as a regulator of strand asymmetry [60, 61]. Indeed, compared with co-expression of green fluorescence protein (GFP), co-expression of LS-encoded CP protein with ncL3 increased the accumulation levels of ncL3 and both polarities of sat-T1 by 23%-67% when combined with the LS replicase, but the asymmetry of sat-T1 replication remained largely unchanged (S1 Fig).

We then investigated whether other viral RNAs have the ability to stimulate sat-T1 replication. To the end, we examined the noncoding RNAs of Fny-CMV and another *cucumovirus*

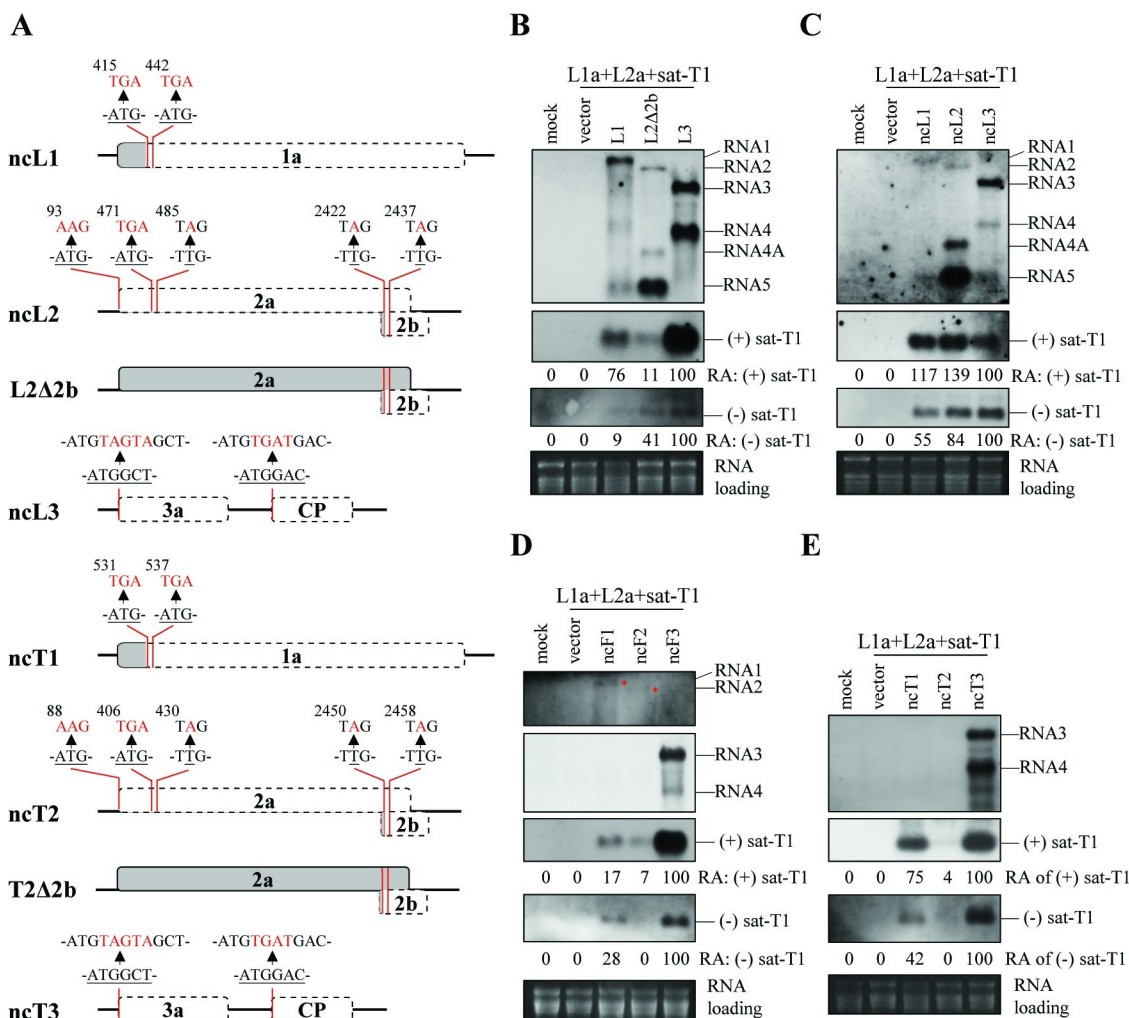

**Fig 2. Viral RNA(s) were essential for LS replication proteins to support satRNA replication, independent of translation products.** (A) Schematic diagrams of the non-coding genomic RNAs of LS-CMV or TAV. All of the mutants were generated using one-step site-directed mutagenesis, with substituted nucleotides highlighted in red. The untranslated regions corresponding to each open reading frame in these mutants are delineated by black dashed lines. (B-E) Northern blotting analyses of the accumulation of sat-T1, viral RNAs or their mutants in the *trans*-replication assays expressing LS replication proteins (L1a and L2a). In the *trans*-replication assays, sat-T1 was co-expressed with RNA1, RNA2Δ2b (lacking 2b expression) and RNA3 of LS-CMV (B) and its non-coding RNAs (ncL1, ncL2 and ncL3) (C), as well as non-coding RNAs of Fny-CMV (ncF1, ncF2, ncF3) (D) or TAV (ncT1, ncT2, ncT3) (E). The *trans*-replication assays were conducted in the 5th true leaves of *Nicotiana benthamiana* plants. Mock plants were treated with the infiltration solution. Vector controls refer as to these samples infiltrated with *Agrobacterium* cells carrying the vector pCB301. Total RNAs were extracted from the infiltrated leaves at 3 days post-infiltration, and subjected to northern blotting hybridization for detection of viral RNAs, their mutants, and positive-sense and negative-sense RNAs of sat-T1. The oligonucleotide probe targeting to the conserved sequence in the 3′ UTR of all CMV strains were used to detect the RNA mutants from LS-CMV (B and C), and RNAs 3 & 4 of Fny-CMV and TAV (D-E). RNAs 1 & 2 of Fny-CMV were detected using both RNA-specific oligonucleotide probe (D). The band intensity corresponding to sat-T1 in each sample was arbitrarily quantified, and the relative accumulation levels were presented below. Ethidium bromide-stained ribosomal RNAs were used to assess the relative loading amounts of the RNA samples.

TAV. Similarly, all the noncoding RNAs of Fny-CMV stimulated sat-T1 replication, albeit with markedly varying efficiency (Fig 2D). Among them, the noncoding RNA3 (ncF3) of Fny-CMV exhibited significantly higher ability to enhance the accumulation of both polarities of sat-T1 compared with the other two noncoding RNAs (ncF1, ncF2) from RNA1 and RNA2. ncF2 displayed the least efficiency in stimulating (+)-stranded sat-T1, with undetectable levels

of (-) strands (Fig 2D). Interestingly, the noncoding RNAs of TAV showed nearly identical patterns to those of Fny-CMV (Fig 2E). In both viruses, the noncoding RNAs from RNA1 and RNA2 were scarcely detected. Overall, these results demonstrate that the stimulation of sat-T1 replication by viral RNAs is not strictly dependent on viral RNA sequences.

We then questioned whether viral RNAs could enhance the replication of other satRNA strains when combined with LS replication proteins. In the *trans*-replication assays with LS replication proteins expressed in *N. benthamiana* plants, each satRNA strain was co-expressed with or without ncL3. Additionally, the expression of satRNAs alone was included as a control. Northern blotting analyses revealed that both polarities of all these satRNAs were detected only when co-expressed with both ncL3 and the replicase of LS-CMV, showing slight variations in their accumulation levels (Fig 3A). Although it was established that the replication proteins of Fny-CMV alone are sufficient to support satRNA replication (Fig 1C), we were intrigued by whether viral RNAs also play a role in satRNA replication when Fny replication proteins are present. Thus, we assessed the accumulation of these seven satRNA strains in the *trans*-replication assays with Fny replicase expressed, along with the addition of ncF3. Northern blotting analyses showed that the presence of ncF3 led to a marked increase in the accumulation levels of both polarities of these satRNA strains, with (+) polarity ranging from 2.9 to 9.3 folds and (-) polarity ranging from 1.7 to 4.7 folds (Fig 3B). Again, all satRNAs were undetectable in the leaves expressing satRNA alone. Taken together, these findings provide compelling evidence to underscore the importance of viral genomic RNAs in enhancing satRNA replication by viral replication proteins, independently of their translation products.

## The TLS element is essential, but not sufficient for viral RNAs to enhance satRNA replication in plants

To gain a deeper understanding of the role of viral RNAs in facilitating satRNA replication, we generated a series of L3 variants and examined their impact on sat-T1 replication using the *trans*-replication assays with the expression of LS replication proteins. Experiments with the L3 mutants, including those lacking MP (L3-ΔMP), CP (L3-ΔCP) or both (L3-ΔMPΔCP), revealed significant findings. Specifically, deletion of either MP or CP resulted in a reduction in the accumulation levels of both RNA3 mutants by 56% and 75%, respectively, compared to the wildtype L3 (Fig 4A & 4B). These deletions substantially reduced the accumulation levels of (-) sat-T1, while having no discernable effect on the levels of (+) sat-T1 (Fig 4A & 4B). Strikingly, unlike L3-ΔMP and L3-ΔCP, L3-ΔMPΔCP was nonviable, failing to prompt the replication of sat-T1 as evidenced by the undetectable levels of both polarities of sat-T1 (Fig 4A & 4B). These results suggest that while the coding sequences of both MP and CP are important for the synthesis of (-) sat-T1, the presence of either one is sufficient for L3 to facilitate sat-T1 replication.

The 3′ UTR of CMV RNAs compromises three distinct regions: a variable region (VR) at the 5′ end, a conserved TLS at the 3′ end, and a highly conserved region (CR) separating them (Figs 4A & S2). To assess the contribution of these domains to RNA3-facilitated sat-T1 replication, we deleted the VR, CR or TLS in ncL3 (Fig 4A). Using the *trans*-replication system expressing LS replication proteins, experiments with the deletion mutants showed that deleting the VR domain had minimal impact on the accumulation of RNA3, while the deletion of the CR or TLS domain reduced the accumulation by 63% or to an undetectable level, respectively, compared to ncL3. When combined with ncL3, ncL3-ΔVR or ncL3-ΔCR, each strand of sat-T1 accumulated at a similarly high level, while being undetectable in combination with ncL3-ΔTLS (Fig 4C). Deletion of both the VR and CR from ncL3 moderately affected the accumulation of the mutant ncL3-ΔVRΔCR and sat-T1, resulting in approximately a 50%

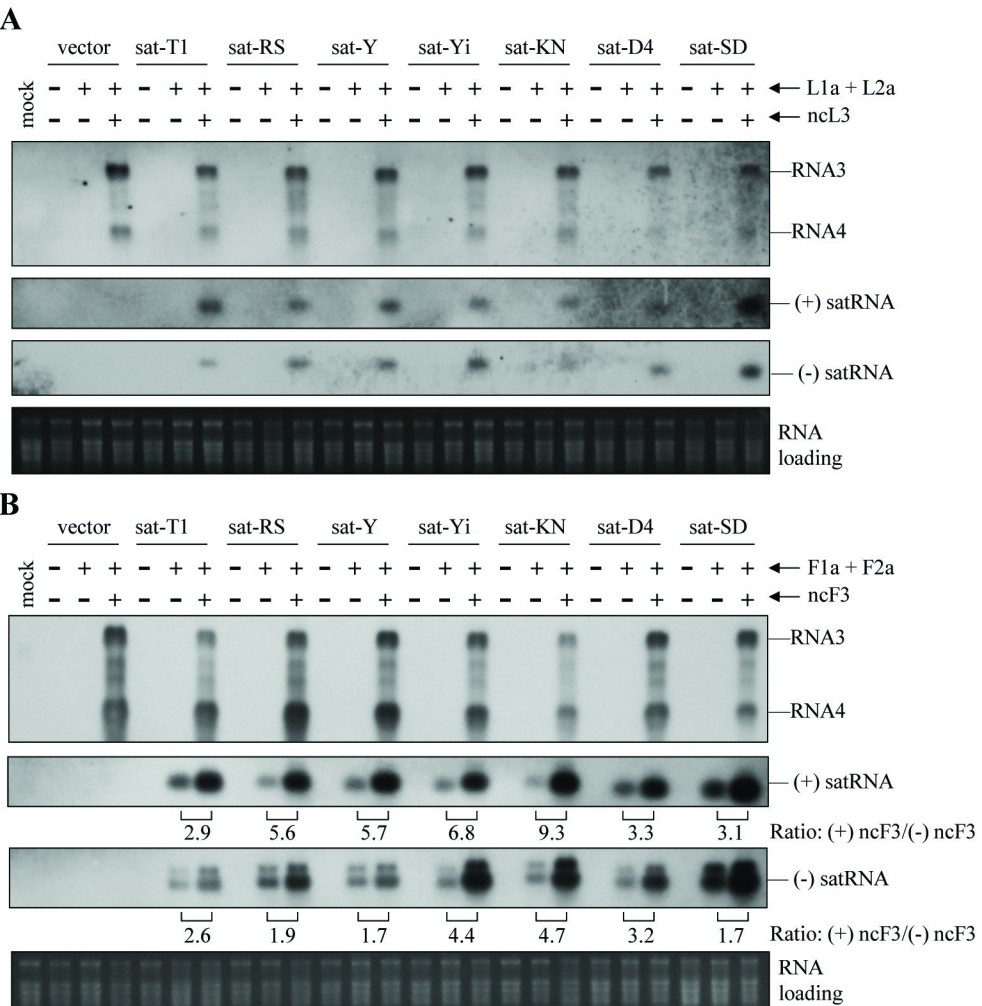

**Fig 3. Replicable viral RNAs enhance satRNA replication in *trans*-replication assays.** Seven satRNA strains were separately expressed alone or co-expressed with the replication proteins from LS-CMV (L1a + L2a) or Fny-CMV (F1a + F2a), together with or without ncL3 (A) or ncF3 (B) in the 5th true leaves of *Nicotiana benthamiana* plants. The RNA silencing suppressor P19 was included in all the treatments. At 3 days post-agroinfiltration, total RNAs were extracted from the infiltrated leaves and subjected to northern blot hybridization for measuring the RNA3 mutants and both positive-sense and negative sense strands of these satRNA strains. Mock plants were treated with infiltration solution alone, while vector controls denote these samples infiltrated with *Agrobacterium* cells carrying the vector pCB301. The ratios of satRNA accumulation in the presence of ncF3 (+) relative to its absence (-) are shown below. Ethidium bromide-stained ribosomal RNAs served as the loading control.

reduction in both (Fig 4D). Providing the TLS *in trans* did not stimulate sat-T1 replication in the replication assay (Fig 4E). Moreover, when the TLS was provided in RNA4 (L4) or its non-coding RNA (ncL4) of LS-CMV, neither viral RNA was effective in stimulating sat-T1 replication, as did the TLS itself (Fig 4E). These results demonstrate the essential role of the TLS, albeit insufficient, for viral RNAs to stimulate sat-T1 replication by LS replicase.

TLS is a functional RNA element commonly utilized by various RNA viruses, exhibiting variations in both sequence and 3′ modifications [62,63]. Well-known viral TLSs include those encoded by BMV, TMV, and TYMV [63,64]. We were intrigued by whether the TLS in CMV viral RNAs could be functionally replaced with TLSs from other viral sources to facilitate satRNA replication. To this end, we precisely replaced the TLS in ncL3 with that of TMV,

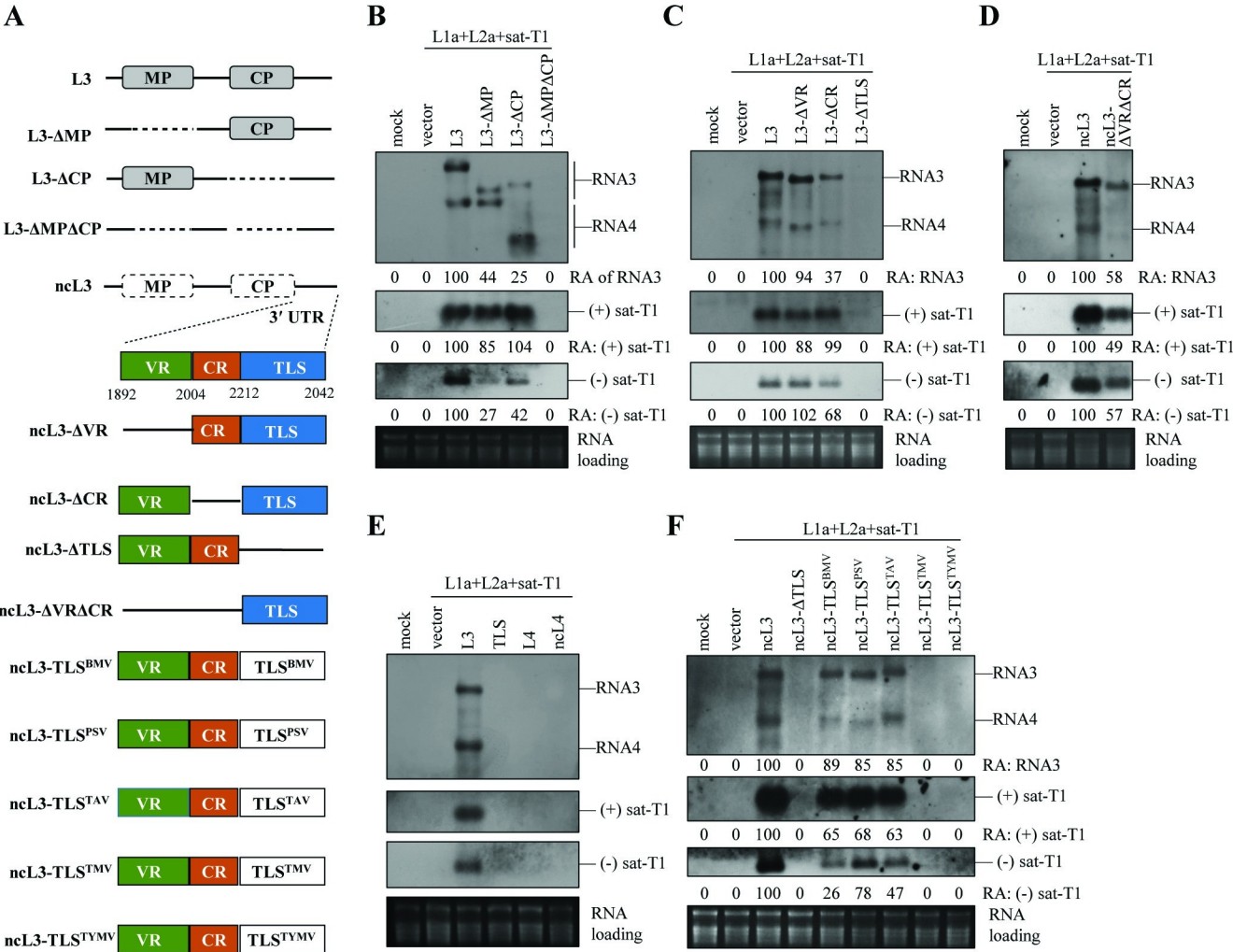

**Fig 4. The TLS element is essential for viral RNAs to enhance satRNA replication in *trans*-replication assays.** (A) Schematic diagrams of L3, ncL3 and their derivatives. The 3′ UTR of CMV RNAs is divided into three regions: a variable region (VR) at the 5′ end, a conserved TLS at the 3′ end, and a highly conserved region (CR) separating them. Deleted sequences in the constructed mutants were indicated by dashed lines. The TLS in ncL3 was substituted with the TLS of BMV, PSV, TAV, TMV, or TYMV, to generate six chimeric ncL3 mutants. (B-D, F) Northern blotting analyses of the accumulation of sat-T1, L3 and its mutants in the 5ᵗʰ true leaves of *Nicotiana benthamiana* transiently expressing LS replication proteins (L1a+L2a) and the RNA silencing suppressor P19. The mutants of L3 or ncL3 tested in these experiments are shown above. (E) Northern blotting analyses of TLS, RNA4 (L4) and its noncoding version (ncL4) of L3, as well as both polarities of sat-T1 in the *trans*-replication assays. In this replication assay, the TLS, L4 and ncL4 were provided separately in *trans* via agroinfiltration. It is worth mentioning that the probe used to detect RNA3 and its subgenomic RNA4 in (C) & (D) is the digoxin-labeled oligonucleotide complementary to the sequence spanning from nt 1200 to 1333 of L3. The digoxin-labeled oligonucleotide probe used to detect these RNAs in (E) is complementary to the sequence positioning at nt 2128–2167 of L3. Mock plants were treated by infiltration solution alone. The relative accumulation levels of RNA3, and positive-sense or negative-sense RNA of sat-T1 in (B-D, F) are shown below. Ethidium bromide-stained ribosomal RNAs served as the loading control.

TYMV, BMV, TAV, or PSV (Figs S3 & 4A), the later three of which belong to the same family *Bromoviridae* as CMV. Subsequently, we tested these chimeric RNAs in *trans*-replication assays, involving the co-expression of sat-T1 and LS replication proteins in *N. benthamiana*. Northern blotting analyses showed that the chimera containing TLS^BMV, TLS^PSV or TLS^TAV accumulated at similar levels to that of ncL3 and enhanced the accumulation of (+) sat-T1, reaching 63%-68% of the ncL3 level. However, the other two chimera were biologically inactive and failed to stimulate the accumulation of both polarities of sat-T1, akin to ncL3-ΔTLS (Fig 4F). These results indicate that the TLS in CMV RNAs can be functionally replaced with

structurally identical TLSs from genetically related viruses, not only supporting the replication of their chimeric RNA3s, but also facilitating satRNA replication.

## Complete replication of viral RNAs is required for enhancing satRNA replication

The TLS of CMV RNA3 contains the promoter essential for initiating the synthesis of (-)-sense RNAs [65]. Consequently, removing the TLS from CMV RNA3 inevitably hampers the production of its (-)-sense RNAs, thereby impeding the generation of its (+)-sense RNAs, as observed in our experiments with the mutant ncL3-ΔTLS (Fig 4C). The indispensable role of TLS for ncL3 in enhancing sat-T1 replication (Fig 4C & 4D) raises the question of whether this necessity hinges solely on its function in the synthesis of (-)-sense RNA3 or together with the synthesis of the coupled (+)-sense RNA3. To this end, we constructed the mutant ncL3-ΔVRΔCRΔ5U, in which, apart from the first nucleotide guanine, the whole 5′ UTR in the L3 derivative ncL3-ΔVRΔCR was replaced with an unrelated sequence, M13R(-48) (Fig 5A). It is worth mentioning that the 5′ UTR contains the essential motif for the synthesis of (+)-sense RNA3 [66]. Additionally, we chose ncL3-ΔVRΔCR as the background to minimize possible interference with the synthesis of (+)-sense RNA3 from both VR and CR. Using the *trans*-replication system, the experiments with the constructed mutant, conducted in parallel with both controls ncL3 and ncL3-ΔVRΔCR, showed that in contrast to both controls, ncL3-ΔVRΔCRΔ5U was undetectable, and could not enhance sat-T1 replication evidenced by both undetectable polarities of sat-T1 (Fig 5B). As anticipated, the (-)-sense RNAs of ncL3-ΔVRΔCRΔ5U were detected by RT-PCR, similar to both controls, but not in the RT-free (RT-) PCR reactions (Fig 5C). This indicates that the TLS served as the promoter to initiate the synthesis of (-)-sense RNAs from the transcribed RNAs of ncL3-ΔVRΔCRΔ5U. Taken together, we suggest that the complete replication of viral RNAs, encompassing synthesis of both polarities, is required for enhanced satRNA replication.

## The recruitment of viral RNAs by viral replicases is the prerequisite to enhancing satRNA replication

To deepen our understanding of the non-template functions of viral RNA in facilitating the multiplication of sat-T1, we attempted to weaken the promoter activity of 5′ UTR or TLS by either deleting a partial sequence of the 5′ UTR or altering the stem-loop C of the TLS in L3, resulting in the generation of L3-Δ5UII and L3-mSLC, respectively (Fig 6A). Meanwhile, we constructed another mutant, L3-ΔBoxB, by deleting the highly conserved Box-B motif in L3 (Fig 6A). Notably, Box-B is a well characterized element in the family *Bromoviridae*, essential for viral RNA recruitment, particularly elucidated in BMV [65,67]. Subsequently, we assessed the impact of co-expressing each mutant on sat-T1 replication using the *trans*-replication assays with the expression of LS replication proteins. Northern blotting analyses showed that, unlike the inactive mutants with complete deletion of the 5′ UTR or the TLS, both L3-Δ5UII and L3-mSLC retained partial replication activity, with accumulation levels reaching 54% or 18% of that observed with L3 (Fig 6B). As expected, the deletion of Box-B proved fatal to L3 replication, rendering the mutant L3-ΔBoxB undetectable (Fig 6B). Interestingly, we observed a similar pattern in the accumulation levels of (+) sat-T1, as did L3 and its mutants, albeit of a distinct pattern in the accumulation of (-) sat-T1 (Fig 6B). These findings suggest that, as one of the non-template functions, the recruitment of viral RNAs acts as a prerequisite for enhancing satRNA replication. Furthermore, it appears that the replication levels of (+) satRNA are positively correlated with the efficiency of viral RNA amplification.

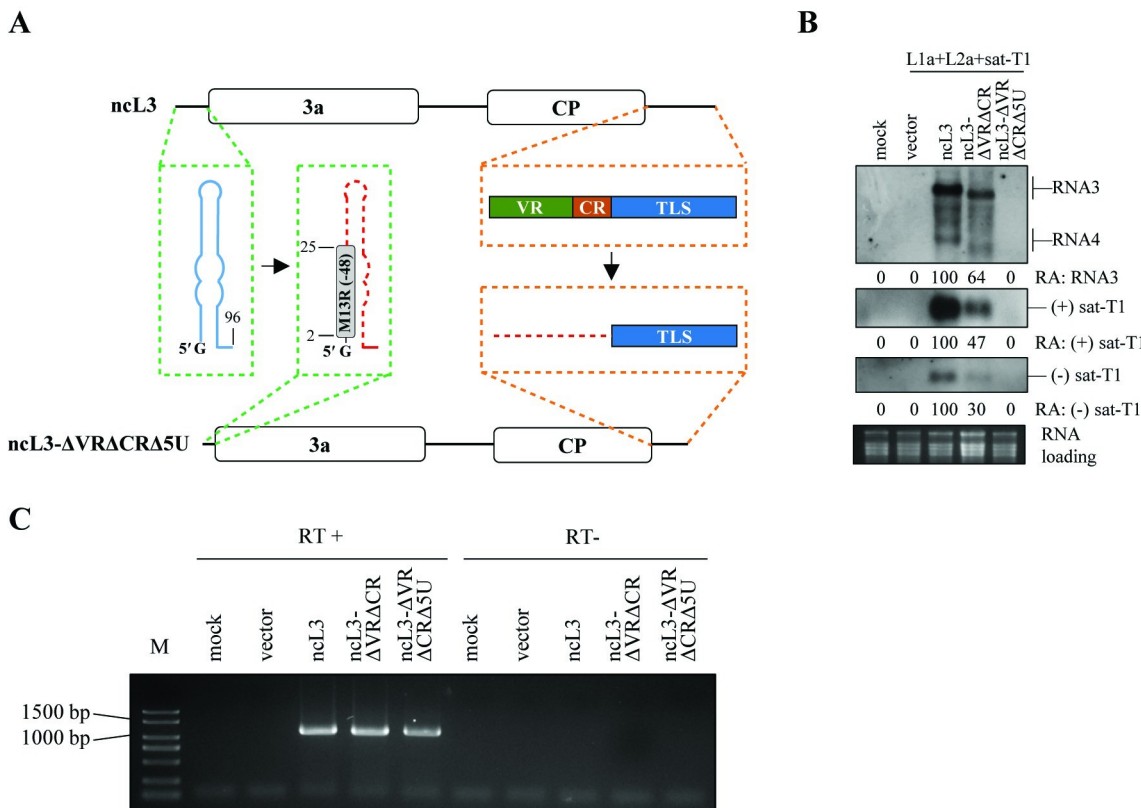

**Fig 5. Complete replication of viral RNAs is required for enhancing satRNA replication in *trans*-replication assays.** (A) Schematic diagrams of ncL3 and its derivative ncL3-ΔVRΔCRΔ5U. The green blocks denote the substitution of the 5′ UTR with the sequence of M13R(-48), and the orange blocks denote the deletion of both VR and CR in the 3′ UTR. (B) Northern blotting analyses of the accumulation of sat-T1, ncL3 and its derivatives ncL3-ΔVRΔCR and ncL3-ΔVRΔCRΔ5U. In the *trans*-replication assay expressing LS replication proteins (L1a+L2a), sat-T1 was co-expressed with ncL3 or its derivatives, as well as the vector (pCB301) control in the 5th true leaves of *Nicotiana benthamiana* plants. At 3 days post-infiltration, total RNAs were extracted from the infiltrated leaves and subjected to northern blot hybridization. The relative accumulation levels of RNA3, and positive-sense or negative sense RNAs of sat-T1 are shown below. Ethidium bromide-stained ribosomal RNAs served as the loading control. (C) Strand-specific RT-PCR for detecting the negative-sense RNAs of ncL3 and its derivative in the RNA samples shown in panel (B). RNA samples were digested with TURBO DNase to remove DNA, and subjected to RT reactions with the primer ncL3a-F. The RT products were amplified using the primer pair ncL3a-F and ncL3b-R. In parallel, total RNAs were amplified directly in PCR reactions without the RT process, indicated with RT(-). PCR products were separated in a 1% agarose gel and observed under UV light after ethidium bromide staining. "M" denotes the DL2000 DNA ladder.

### The 1a protein of LS-CMV is partially responsible for the defect of LS replicase in supporting satRNA replication in the absence of viral RNAs

The finding that LS replication proteins were defective in multiplying satRNAs prompted us to determine which viral replication protein of LS-CMV was responsible for this defect. To delve deeper into this matter, we conducted an experiment involving the exchange of the replicase components between Fny and LS. We then assessed the performance of both the heterologous and parental replication proteins in replicating L3, sat-T1, or both together in *N. benthamiana* plants using the *trans*-replication system. In the case of L3 replication alone, when combined with L1a, L2a and F2a exhibited nearly identical abilities to replicate L3, which was 87% lower compared with the parental replication proteins F1a + F2a. When combined with L2a, the replacement of L1a with F1a led to an over 3-fold increase in the replication of L3, but still not as efficient as F1a + F2a (Fig 7A). These results demonstrate that both heterologous replicases were biologically compatible at least in replicating viral RNA. In the case of sat-T1 replication

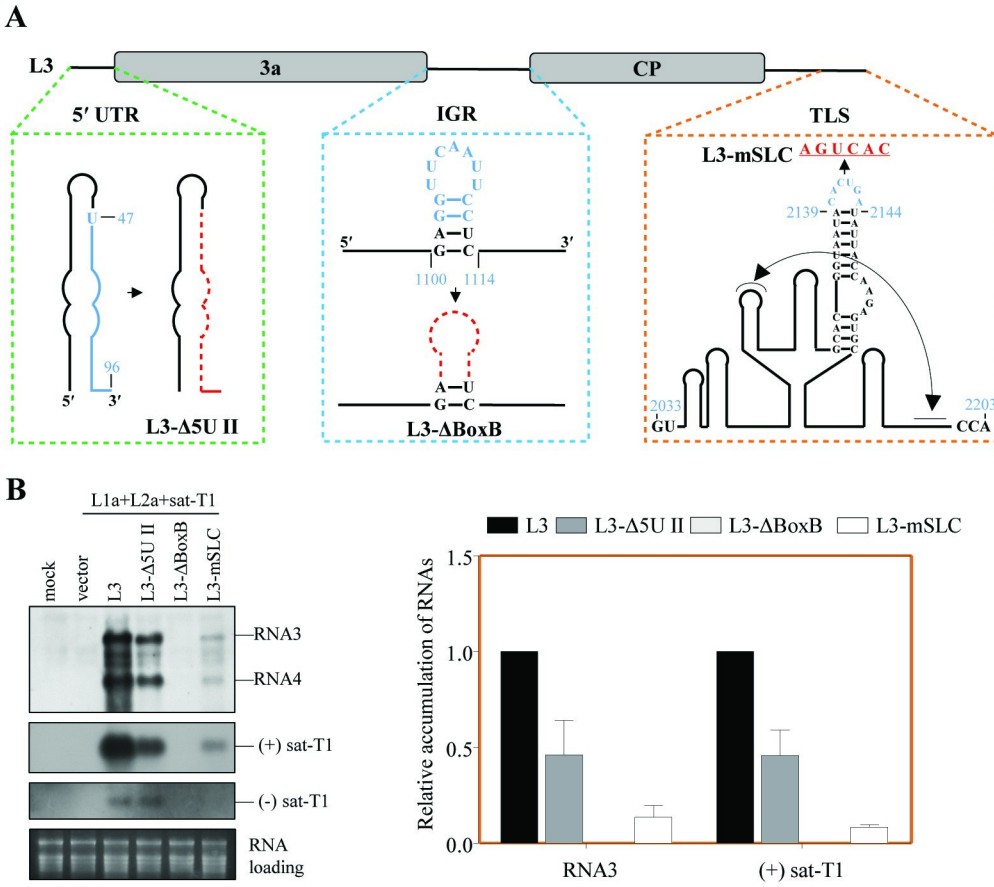

**Fig 6. The recruitment element Box-B is indispensable for viral RNAs to enhance satRNA replication.** (A) Schematic diagrams of L3 and its derivatives. The sequence from positions 47–96 nt in the 5′ UTR was removed from L3, to create L3-Δ5UII, as shown in the rectangle with green dashed lines. The Box-B motif in the IGR was removed from L3 to generate the mutant L3-ΔBoxB, as shown in the rectangle with blue dashed lines. The stem-loop C in the TLS was mutated with the nucleotides (AGUCAC) to create the mutant L3-mSLC, as shown in the rectangle with brown dashed lines. (B) Northern blotting analyses of the accumulation of RNA3 and sat-T1 in the presence of LS replication proteins (L1a+L2a). In the *trans*-replication assays, sat-T1 was co-expressed with L3, the aforementioned mutants, or the vector pCB301 in the 5th true leaves of *Nicotiana benthamiana* plants. At 3 days post-infiltration, total RNAs were extracted from the infiltrated leaves and subjected to northern blot hybridization. Mock plants were treated by infiltration solution alone. Ethidium bromide-stained ribosomal RNAs were used to ensure the equal loading for all RNA samples. The relative accumulation levels of sat-T1 or RNA3 are depicted in the graph on the right. Mean values with standard errors were calculated from three independent biological experiments.

alone, the heterologous recombinant (L1a + F2a) displayed defects in replicating sat-T1, mirroring the performance of LS replication proteins (L1a + L2a) (Fig 7B). The other heterologous recombinant (F1a + L2a) exhibited a certain capability to stimulate the replication of sat-T1, with the accumulation of (+) sat-T1 reaching only 6% of that stimulated by F1a + F2a (Fig 7B). Notably, sat-T1 was undetectable in the presence of either F1a or F2a alone (Fig 7B). When L3 was co-expressed with sat-T1, the heterologous combination F1a + L2a was as closely efficient as F1a + F2a in amplifying (+) sat-T1, albeit of much less efficiency in building up the (-) strands. Compared with L1a + L2a, the heterologous replicase L1a + F2a was even less efficient in amplifying (+)-strands of sat-T1, while exhibiting a similar ability to synthesize (-) strands (Fig 7C). These results indicate both heterologous replicases were adaptable in replicating viral RNAs, including satRNAs when viral RNAs are present, despite the varied efficiencies. Taken

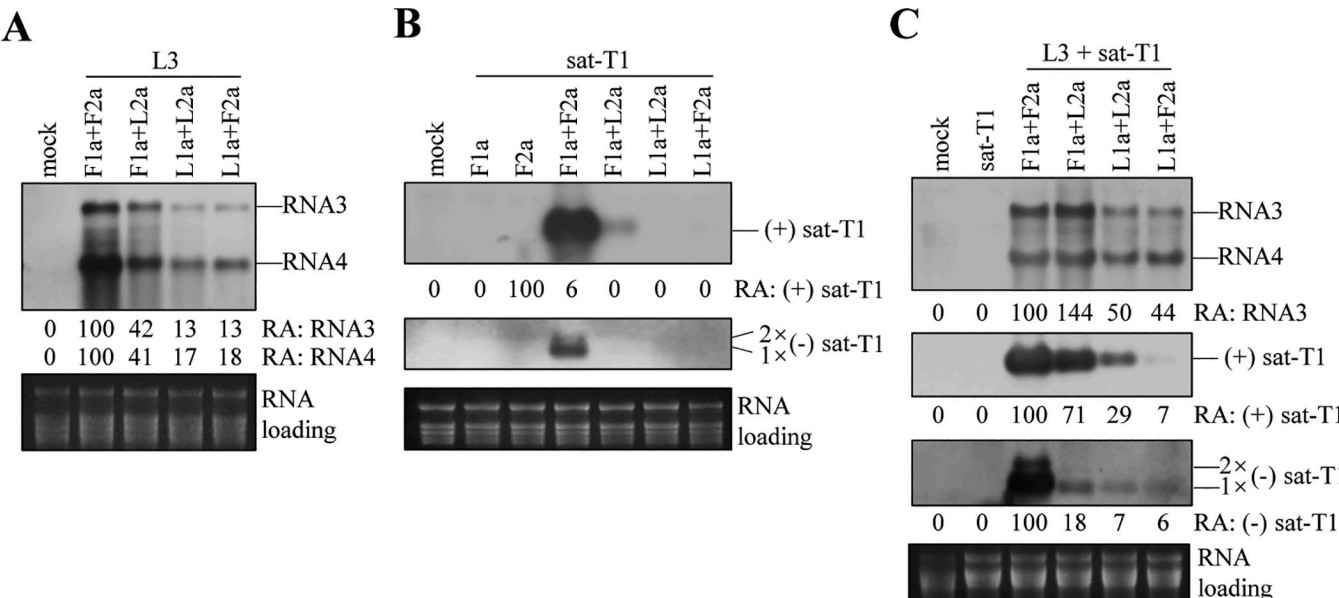

**Fig 7. Both replicase components of LS-CMV are responsible for the defect in supporting satRNA replication.** (A-C) Northern blotting analyses of the accumulation of L3 (A), sat-T1 (B), or both together (C). L3, sat-T1, or both were co-expressed with the replication proteins of Fny or LS, or their heterologous recombinants (F1a + L2a, L1a + F2a) in the 5$^{th}$ true leaves of *Nicotiana benthamiana* plants. Co-expression of sat-T1 with F1a or F2a served as negative controls in (B). At 3 days post-infiltration, total RNAs were extracted from the infiltrated leaves and subjected to northern blot hybridization. Mock plants were treated by infiltration solution alone. The relative accumulation levels of L3, and positive-sense or negative sense RNAs of sat-T1 were presented below. Ethidium bromide-stained ribosomal RNAs were used to assess the relative loading amounts of the RNA samples.

together, the 1a protein of LS-CMV was partially responsible for the defect of LS replicase in supporting sat-T1 multiplication in plants, when viral RNAs were absent.

## LS replicase is not as proficient as that of Fny-CMV in recruiting (+)-sense RNAs of sat-T1

CMV 1a protein is responsible for the recruitment of viral RNAs into VROs. Thus, our finding that the failure of LS replicases to proliferate sat-T1 was partially attributed to the 1a protein raised the question of whether LS replicase was deficient in recruiting satRNAs. To the end, we attempted to develop a method for examining the recruitment of satRNAs by viral replicases. The methodology is based on the functional replacement of the Box-B motif with satRNA sequences within a modified RNA3. Considering the robust capability of LS replication proteins in replicating RNA3 (F3) of Fny-CMV (Fig 2C), we opted to select F3 as the replication template to incorporate mutations of the Box-B or foreign sequences (Fig 8A). Expectedly, mutating the Box-B in F3 resulted in a nearly complete abolition of replication for this mutant, denoted as mF3, when combined with LS replication proteins (Fig 8B). This finding strongly suggests that the mutation of the Box-B effectively hindered the recruitment of F3 by LS replicase.

Initially, we found that substitution of the Box-B sequence in either F3 or L3 with (+)-strand of sat-T1 was detrimental to the variants of both RNA3s when combined with the replicase of LS-CMV (S4 Fig). Subsequently, we attempted to incorporate either the (+)- or (-)-strand of sat-T1 downstream of the CP in mF3, yielding mF3-T1(+) and mF3-T1(-), respectively (Fig 8A). In parallel, a portion of the GUS gene equivalent in size of sat-T1 was integrated downstream of the CP in F3 or mF3, resulting in the creation of F3-gus or mF3-gus, respectively, serving as controls (Fig 8A). We then proceeded to assess the accumulation levels

of these F3 variants in the presence of the replication proteins of Fny or LS in *N. benthamiana*. Northern blotting analyses yielded expected results for both controls: F3-gus was replicated substantially, while mF3-gus was undetectable in both experimental setups (Fig 8C). Interestingly, mF3-T1(+) exhibited distinctly different responses to the replication proteins of Fny and LS. mF3-T1(+) was weakly detected in the presence of LS replication proteins, whereas when combined with Fny replication proteins, it accumulated significantly, reaching up to 68% of the level observed with F3-gus (Fig 8C). Unexpectedly, the insertion of (-)-strand of sat-T1 [mF3-T1(-)] substantially increased the accumulation of mF3 to 49% or 65% of the level of F3-gus, in the presence of the replication proteins of LS-CMV or Fny-CMV, respectively. Collectively, the data obtained from the use of the developed method provided evidence in support of our hypothesis that, unlike Fny replicase, LS replicase exhibited a diminished capability to recruit satRNAs for replication in plants.

## Differential ability of the 1a proteins of LS-CMV and Fny-CMV in binding sat-T1 in plants

To gain deeper insight into the differential ability of the replicases of both CMV strains in recruiting satRNAs, we endeavored to develop a second method capable of examining the

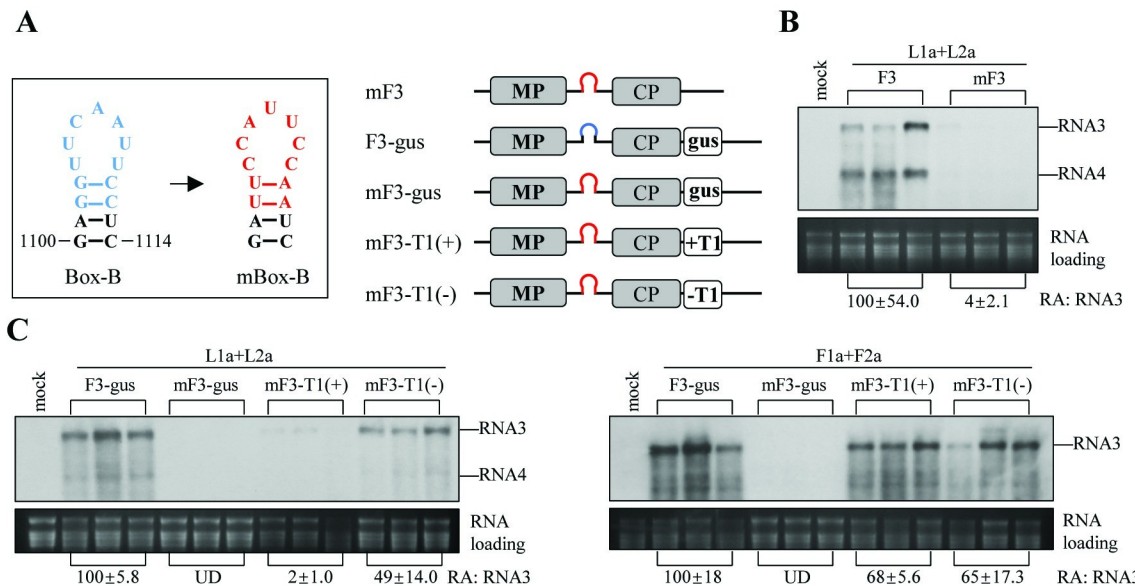

**Fig 8. The replication proteins of Fny-CMV and LS-CMV exhibit significant differences in their ability to recruit positive-sense RNA of sat-T1.** (A) The hairpin structures containing the Box-B sequence in blue and the mutated Box-B (mBox-B) in red. mF3 denotes the F3 mutant, in which the Box-B was substituted with mBox-B. mF3-T1(+) and mF3-T1(-) are mF3 derivatives with positive-sense or negative-sense RNA of sat-T1 inserted between the CP and 3′ UTR in mF3, respectively. A fragment of the GUS gene (337 nt), equivalent in size of sat-T1, was introduced into F3 or mF3, resulting to the creation of F3-gus or mF3-gus, respectively. (B) The accumulation levels of F3 and mF3 in the *trans*-replication assay. Either F3 or mF3 was co-expressed with LS replication proteins and P19 in the 5[th] true leaves of *Nicotiana benthamiana* plants. Mock plants were treated with infiltration solution. At 3 days post-infiltration, total RNAs were extracted separately from three infiltrated leaves for each treatment and subjected to northern blotting analyses. The relative accumulation levels of F3 and mF3 are shown below as the mean values with standard errors from three independent biological samples. (C) Determination of the replication activities of mF3-T1(+), mF3-T1(-), and the controls F3-gus and mF3-gus. These four F3 derivatives was co-expressed with the P19 suppressor and the replication proteins of LS-CMV (upper panel) or Fny-CMV (the lower panel) in the 5[th] true leaves of *N. benthamiana* plants. Mock plants were treated with infiltration solution. At 3 days post-infiltration, total RNAs were extracted separately from three infiltrated leaves for each treatment, and analyzed by northern blot hybridization. The relative accumulation levels with standard errors shown below were calculated from three independent biological samples. "UD" denotes the undetectable level. Ethidium bromide-stained ribosomal RNAs were used as a loading control for normalization of the relative accumulation levels.

molecular interaction of sat-T1 with CMV 1a protein. In this developed method, both L1a and F1a were fused with mCherry, which produced red fluorescence exclusively in the cytoplasm, with punctate spots outlining the cell (Fig 9B), when transiently expressed in plant epidemic cells. This subcellular localization mirrors that of CMV 1a protein as reported previously [38]. The bacteriophage MS2 CP (MCP) was fused with YPF, followed by an NLS sequence at the carboxyl terminus (Fig 9A). The fusion protein MCP-YFPnls was enriched solely in nuclei when transiently expressed in the epidermal cells (Fig 9B). As the binding site of MCP, six repeats of the MS2 stem-loop (6×MS2) were linked to the 5′ end of sat-T1 or a GUS fragment, creating the two chimeric RNAs, 6×MS2-satT1 and 6×MS2-gus, respectively (Fig 9A). When the chimeric RNAs were individually co-expressed with MCP-YFPnls and either F1a-mCherry or L1a-mCherry in the leaf tissues of *N. benthamiana*. Interestingly, we observed nuclear localization of F1a-mCherry only in the 6×MS2-satT1-expressing leaf samples, not in the 6×MS2-gus-expressing ones (Fig 9C). However, in the case of L1a-mCherry, such nuclear localization was not observed regardless of whether 6×MS2-satT1 or 6×MS2-gus was present (Fig 9C). Furthermore, no alteration to the subcellular localization of MCP-YFPnls was observed in all these circumstances. When MCP-YFPnls was absent, co-expression of F1a-mCherry with 6×MS2-satT1 did not result in the nuclear accumulation of F1a-mCherry (S5 Fig). These results suggest that F1a-mCherry possesses the ability to bind sat-T1, leading to its accumulation in nuclei.

## Discussion

In the work, we investigated the involvement of the non-template functions of helper virus RNAs in satRNA replication. Unlike the *in vitro* competition model reported previously [52], our findings unequivocally demonstrate that replicable RNAs of CMV enhance satRNA proliferation in *trans*-replication experiments. This enhancement is suggestively contingent upon the recruitment and complete replication of helper virus RNAs. Thus, our study provides experimental evidence to reveal the vital role of the non-template functions of CMV RNAs in enhancing satRNA proliferation in plants. These findings prompt us to re-think the interplay between helper virus RNAs and satRNAs in their replication. Furthermore, our work would shed some light on the spatial association of satRNAs with helper virus RNAs in their replication.

It was established four decades ago that, in the absence of their helper virus, satRNAs can survive for up to 25 days in plants [68, 69]. Notably, CMV satRNA was discovered to possess the ability to localize in cell nuclei independently of its helper virus [70], which is facilitated by interaction with the host bromodomain-containing protein [9]. Significantly, nuclear-localized satRNAs may exploit host RNA polymerases to generate both monomers and multimers of (+)- and (-)-strands, which are detectable through RNA gel blotting [70,71]. These findings offer a mechanistic explanation for the helper virus-independent, long-term survival of CMV satRNAs in plants. However, in our research, we did not detect either (+)- or (-)-strands of satRNAs in the leaves expressing satRNAs alone. A similar scenario with sat-T1 and its lethal mutants was observed in our previous work [51]. Notably, our methodology was consistently unable to detect both polar strands of satRNAs in the absence of CMV. Alternatively, both polar strands of satRNAs detected in the presence of CMV replicases are the outcome of helper virus-dependent satRNA replication. When comparing our methodology with the one reported to successfully detect helper virus-free satRNAs [70], several important differences emerge, including probe labeling (Digoxin vs. isotope $P^{32}$), RNA amount loaded (2 μg vs. 20 μg), and plant species (*N. benthamiana* vs. *N. clevelandii*). These differences provide a reasonable explanation for the discrepancy in satRNA detection in the absence of CMV.

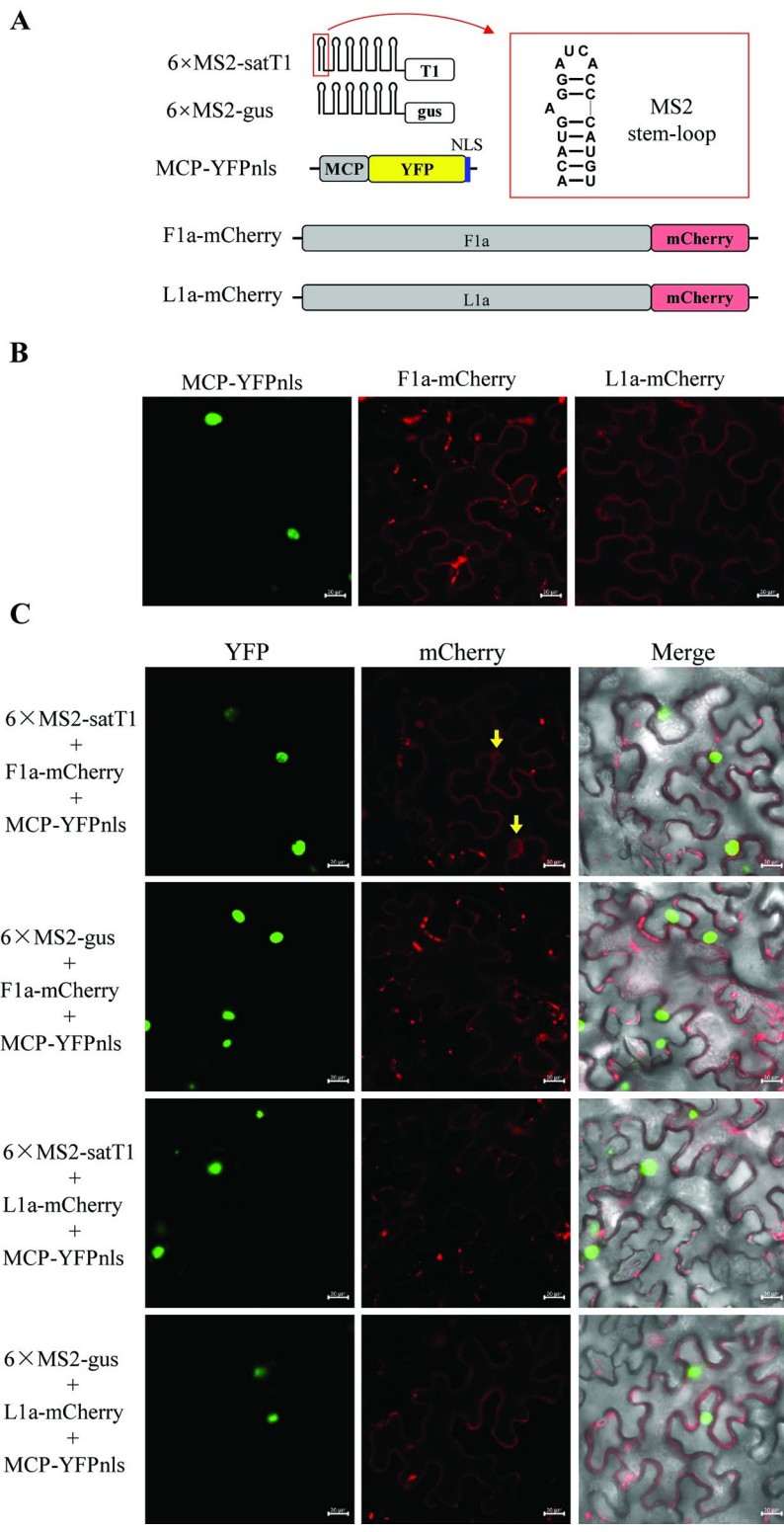

**Fig 9. The 1a protein of Fny-CMV, but not LS-CMV, displays co-localization in nuclei with MCP-YFPnls.** (A) Schematic diagrams of the DNA constructs used in the developed method for analyzing the interaction of CMV 1a with sat-T1. The bacteriophage MS2 stem-loop structure containing the binding site of MS2 CP (MCP), as shown in the rectangle with red dashed lines. Six copies of MS2 stem-loop (6×MS2) were linked at the 5′ end of sat-T1 or an equal-size fragment of the Gus gene, to create 6×MS2-satT1 and 6×MS2-Gus, respectively. The coding sequence of

MCP was fused with YFP, followed by a nuclear localization signal, generating MCP-YFPnls. F1a and L1a were tagged with a copy of mCherry at their C terminus. (B) The subcellular localization of mCherry-tagged 1a proteins and YFPnls-tagged MCP. These proteins were individually expressed with p19 in the leaves of *Nicotiana benthamiana* plants, and subjected to fluorescence visualization using a laser confocal microscopy at 2 days post-agroinfiltration (DAPI). (C) Subcellular distributions of F1a-mCherry and L1a-mCherry when co-expressed with MCP-YFPnls and either 6×MS2-satT1 or 6×MS2-Gus. The 6$^{th}$ true leaves of 3-weeks old *N. benthamiana* plants were infiltrated with the mixture of *Agrobacterium* cells to express the fusion proteins and RNAs as indicated. Fluorescence was visualized at 2 DAPI. The green color represents the fluorescence signal omitted from the fusion protein MCP-YFPnls, and the red color indicates the signals from either F1a-mCherry or L1a-mCherry. Two arrows in yellow indicate the nuclear accumulation of F1a-mCherry, when co-expressed with MCP-YFPnls and 6×MS2-satT1, but not with 6×MS2-Gus. The scale bars denote 20 μm.

Our findings demonstrate that, independent of their translation products, helper virus RNAs have additional abilities to stimulate satRNA replication. A similar scenario was previously reported regarding the tobacco ringspot virus-associated endogenous satRNA, which is latently present, but can be stimulated by two non-accumulating satRNA mutants [72]. However, it is still unknown how the non-accumulating satRNA mutants function in this event. Our data demonstrates that CMV RNAs rely on their recruitment and replication for the stimulated replication of satRNAs (Figs 5–6 & 8–9). In addition, plant hosts play a vital role as well in the replication levels of satRNAs, as demonstrated by early studies showing that CMV satRNAs accumulate at limited levels in squash but reach high levels in tobacco [73]. Moreover, it was found that the existence of satRNAs in a major proportion of field CMV isolates is only detectable after passaging them in tobacco. Thus, trace amounts of CMV satRNAs can be amplified under specific circumstances. Combined with our findings, this suggests that CMV RNAs cooperate together with host factors to determine the replication level of satRNAs in host plants.

Viral RNAs serve as scaffolds, binding viral replicases and host proteins to facilitate the assembly of VRCs [31–33]. In situations where viral RNA-binding host proteins are absent, the assembled VRCs become functionally impaired, leading to a decrease in replication efficiency [31]. Our data suggest that VRCs assembled with satRNAs are less efficient in amplifying satRNAs compared to those assembled with CMV RNAs. This could be attributed to the absence of specific RNA elements in satRNAs, which are typically present in helper virus RNAs for binding replication-related host factors. As a result, we propose a plausible molecular model for the non-template functions of viral RNAs in stimulating satRNA replication (Fig 10). As reported recently [38], when viral RNAs are absent, CMV replicases themselves can establish viral RNA-free VRO-like structures in plant cells. Such VRO-like structures are believed to be effective in replicating satRNAs once satRNAs are recruited by CMV 1a protein, just as does the replicase of Fny-CMV (Figs 1 & 3). The proposed model suggests that VROs generated by helper virus RNAs possess high replication activity, capable of hosting a portion of satRNA molecules for replication, thereby enhancing replication of satRNAs. Our results indicate that all CMV genomic RNAs can boost satRNA replication, implying that satRNAs can make use of the VROs created by any helper virus genomic RNA. Thus, our findings shed some light on the spatial association in VROs governing the replication of helper virus RNAs and satRNAs.

Interestingly, we found that the replicase of LS-CMV is defective at replicating satRNAs when viral RNAs are absent. Our data hints two potential causes for this defect. One is the weak replication activity, as evidenced by the results showing that the replicase of LS-CMV is not as proficient as the replicase of Fny-CMV in amplifying RNA3s of both CMV strains (Figs 1&7). The second is the poor capability of recruiting the (+)-strand of satRNAs (Fig 8). These causes align with our finding that both the 1a and 2a proteins of Fny-CMV contribute separately to the efficiency of both RNA3 replication and satRNA replication (Fig 7) because CMV 1a and 2a are involved in RNA recruitment and RNA synthesis, respectively. The difference in

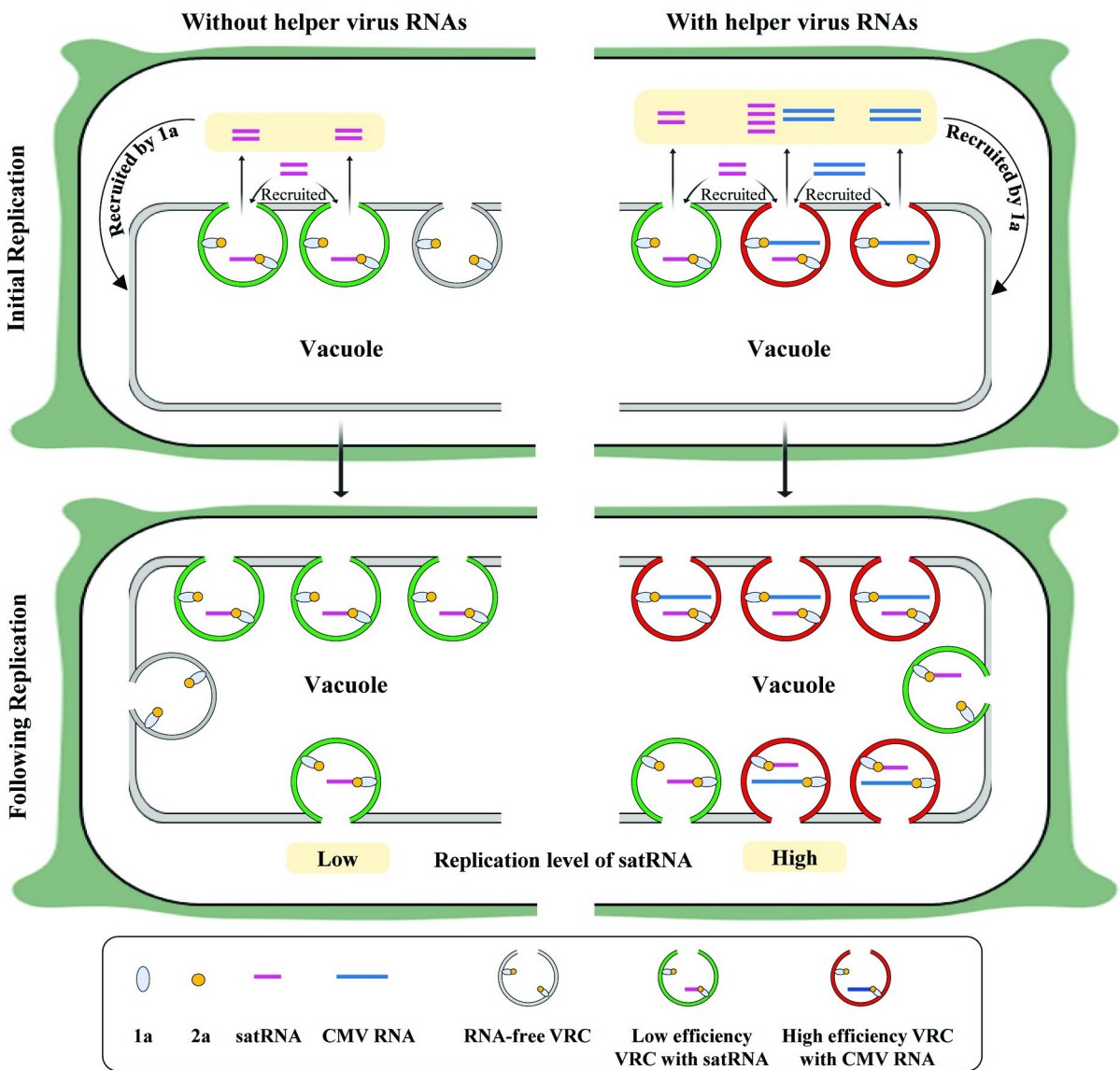

**Fig 10. A proposed model of satRNA replication stimulated by viral RNAs in plants.** During the initial replication stage, CMV replication proteins localize to tonoplast and remodel it to create viral replication organelles (VROs). This process involves the recruitment of satRNAs, viral RNAs, or both to assemble viral replication complexes (VRCs). Notably, VROs could be free of both CMV and satellite RNAs, as viral replication proteins themselves can form VRO-like spherules, as reported previously [38]. VRCs assembled with satRNAs exhibit lower replication activity (indicated by VROs enclosed in green circles), whereas those formed with viral RNAs exhibit high activity (indicated by VROs enclosed in red circles). These highly active VRCs replicate not only viral RNAs, but also satRNAs when satRNAs are recruited alongside viral RNAs into the same VROs. However, considering that LS replication proteins have limited capability for satRNA recruitment, some VROs may lack satRNAs (indicated by VROs enclosed in gray circles) in the absence of viral RNAs. Consequently, VRCs formed in the presence of satRNAs alone would produce less satRNAs compared to those formed in the presence of both viral and satellite RNAs. Following the initial replication, more satRNAs, along with viral RNAs produced during the initial replication, participate in the creation of new VROs. This is expected to contribute to the enhanced proliferation of satRNAs with the assistance of viral RNAs.

replication efficiency is reasonably acceptable due to low sequence identity, ranging from 76.7–78.2% for the 1a ORFs and 69.4–71.3% for the 2a ORFs between CMV strains of subgroups I and II [34]. LS-CMV has been reported to have a preference for supporting satRNA infection [74]. This preference hinges on a 10-nucleotide difference within nucleotides 191–279 of satRNAs [74], where the sequence forms a biologically important ϒ-shaped structure

[51]. This Υ-shaped structure incorporates a pseudoknot, which is crucial for satRNA viability in CMV-infected plants [51]. Considering that the pseudoknot in the 3′ UTRs of viral genomic RNAs is involved in viral replication by interplaying with viral replicase [62,75], it would be interesting to investigate the interaction between the Υ-shaped RNA structure element and the replicase of the LS or Fny strain. Such studies might help for better understanding the differential ability of CMV replicase to support satRNA replication.

In addition, we observed a significant disparity between the replicases of Fny-CMV and LS-CMV in producing (-)-stranded dimers of satRNAs. While (-)-stranded dimers were substantially detected in all tested satRNA isolates when combined with the replicase of Fny-CMV, they were barely detectable in the circumstance of LS replicase, even in the presence of LS-CMV infection (Figs 1 & 3). This suggests that CMV replicase is the key determinant in the production of (-)-stranded dimers of satRNAs. Q-CMV, belonging to the same subgroup as LS-CMV, exhibits a high efficiency in synthesizing (-)-stranded dimers of its cognate satellite Q-sat in infected *N. benthamiana* plants [70]. Thus, it would be interesting to determine the key residue(s) responsible for the production of (-)-stranded dimers by examining the differential residues between the replicases of the LS and Q strains. Experiments with the functional replacement of Box-B in mF3 with sat-T1 strongly suggest that (-)-strand of sat-T1 can be effectively recruited by both replicases of Fny-CMV and LS-CMV (Fig 8). The (-)-strands of monomers and multimers of satRNAs can serve as templates for CMV to initiate satRNA replication [71,76]. Thus, the (-)-stranded dimers of satRNAs produced by the replicase of Fny-CMV would positively feedback to enhance satRNA replication. This could provide another explanation for the distinction between the replicases of Fny-CMV and LS-CMV in the replication efficiency of satRNAs.

In summary, our work presents experimental evidence highlighting the significance of the non-template functions of viral RNAs in satRNA replication within plants. It is evident that both the recruitment and full replication of viral RNAs play pivotal roles in boosting satRNA amplification. We believe this work will prompt re-evaluation of the prevailing consensus that HV RNAs are antagonistic against satRNA replication by competing for limited viral and host resources. Instead, our findings suggest a more nuanced relationship, emphasizing the intricate interplay between satRNAs and cognate helper viruses.

## Supporting information

**S1 Fig. *Trans*-provision of CP increased the accumulation levels of non-coding RNA3 and sat-T1 amplified by the replicase of LS-CMV.** The 5th true leaves of *N. benthamiana* plants were infiltrated with *Agrobacterium* cells to express the replicase (L1a + L2a) and its noncoding RNA3 (ncL3) of LS-CMV, sat-T1, and either GFP or the CP protein of LS-CMV. In parallel, co-expression of the viral replicase and sat-T1 was used as the negative control. At 3 days post-agroinfiltration, total RNAs were extracted from the infiltrated leaves and subjected for RNA gel blotting analyses of ncRNA3, RNA4, and both polarities of sat-T1. The relative accumulation levels of these RNAs are shown below. Ethidium bromide-stained ribosomal RNAs were used for assessing the loading amounts of the RNA samples.
(TIF)

**S2 Fig. Sequence alignment of the 3′ UTRs from the genomic RNAs of Fny-CMV and LS-CMV.** The sequence alignment was carried out using the MegAlign program in DNAstar. The 3′ UTR sequence is divided into three regions: a variable region (VR) at the 5′ end, a conserved tRNA-like structure (TLS) at the 3′ end, and a highly conserved stretch region (CR) separating them.
(TIF)

**S3 Fig. RNA structures of the TLS elements from six RNA viruses.** (A) TLS$^{BMV}$: the TLS from brome mosaic virus (BMV) RNA3, (B) TLS$^{PSV}$: the TLS from peanut stunt virus (PSV) RNA3. (C) TLS$^{TAV}$: the TLS from tomato aspermy virus (TAV) RNA3. (D) TLS$^{TMV}$: the TLS from tobacco mosaic virus (TMV). (E) TLS$^{TYMV}$: the TLS from turnip yellow mosaic virus (TYMV). (F) TLS$^{CMV}$: the TLS from RNA3 of cucumber mosaic virus LS strain. The RNA structures of TLS$^{BMV}$, TLS$^{TYMV}$ and TLS$^{TMV}$ were reported previously [63–64]. The RNA structures of TLS$^{PSV}$, TLS$^{TAV}$ and TLS$^{CMV}$ were predicted based on that of TLS$^{BMV}$. All these structures were redrawn using RNA2Drawer (available at https://rna2drawer.app/). The dash lines with arrows indicate pseudoknots formed by the base-paring of the nucleotide sequences colored red.
(TIF)

**S4 Fig. Substitution of the Box-B motif in RNA3 with sat-T1 inactivated the replication of the RNA3 variants by the replicase of LS-CMV.** (A) Schematic diagrams of RNA3 and its derivative (R3-ΔBox-T1), in which the Box-B sequence was substituted with the (+) strand of sat-T1. (B) Northern blotting analyses of the accumulation of RNA3 and its variants. RNA3 from Fny-CMV (F3) or LS-CMV (L3), as well as their mutants or vector pCB301 was separately co-expressed with the replicase of LS-CMV, together with the RNA silencing suppressor P19. At 3 days post-agroin-filtration, total RNAs were extracted from the infiltrated leaves and subjected to northern blot hybridization. Mock plants were treated by infiltration solution alone. Ethidium bromide-stained ribosomal RNAs were used to assess the loading amounts of all RNA samples.
(TIF)

**S5 Fig. Subcellular distribution of F1a-mCherry when co-expressed with 6×MS2-satT1 in the leaf tissues of *Nicotiana benthamiana*.** The lower epidermis of the leaves was infiltrated with *Agrobacterium* cells harboring the binary plasmids to express F1a-mCherry, 6×MS2-satT1, and p19. At 2 days post-agroinfiltration, the infiltrated leaves were subjected to Laser confocal microscopy for visualizing red fluorescence omitted from F1a-mCherry.
(TIF)

**S1 Table. Primers used for constructing plasmids and strand-specific reverse transcription-polymerase chain reactions.**
(PDF)

**S1 Excel. Minimal Set Data used to calculate the mean values and standard errors in Fig 6B and Figs 8C.**
(XLSX)

**S2 Excel. Minimal Set Data for Grayscale values corresponding to bands shown in all gel blots.**
(XLSX)

## Acknowledgments

We thank Professor Peter Palukaitis for providing the T7 promoter-based infectious clones of Fny-CMV and LS-CMV, and Professor Huishan Guo for providing the plasmid 35S-sat-SD. We thank Ms Yunqin Li for her kind help in operating the confocal microscope.

## Author Contributions

**Conceptualization:** Qiansheng Liao, Zhiyou Du.

**Data curation:** Zhiyou Du.

**Formal analysis:** Zimu Qiao, Jin Wang, Kaiyun Huang, Honghao Hu, Zhouhang Gu, Qiansheng Liao, Zhiyou Du.

**Funding acquisition:** Zhiyou Du.

**Methodology:** Zimu Qiao, Jin Wang, Kaiyun Huang, Honghao Hu.

**Supervision:** Zhiyou Du.

**Validation:** Zhouhang Gu, Zhiyou Du.

**Writing – original draft:** Zimu Qiao, Kaiyun Huang, Zhiyou Du.

**Writing – review & editing:** Zhiyou Du.

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
