## [Decision Letter · Decision Letter 0]

12 Nov 2023

Dear Dr. Du,

We are grateful to you for submitting your manuscript "The non-template functions of helper virus RNAs create optimal replication conditions to enhance the proliferation of satellite RNAs" (PPATHOGENS-D-23-01799) for review by PLOS Pathogens. As with all papers submitted to the journal, your manuscript was carefully reviewed by an Academic Editor in consultation with the Editorial Board. The manuscript was also evaluated by three independent reviewers. The reviewers raised several concerns regarding the experimental rigor. Based on the reviews, we recommend major revision of the manuscript.

A major concern is that the authors should consider the possibility of sat-RNA long-term survival independent of helper virus. Reviewer 3 proposed several experiments that could help to clarify the situation. The current experimental design does not appear to be able to eliminate several possibilities that do not support the main conclusions. These possibilities should be revisited by incorporating several essential controls as suggested by the reviewers.

We cannot make any decision about publication until we have seen the revised manuscript and your response to the reviewers' comments. Your revised manuscript is also likely to be sent to reviewers for further evaluation.

Sincerely,

Ying Wang, Ph.D.

Academic Editor

PLOS Pathogens

Shou-Wei Ding

Section Editor

PLOS Pathogens

Kasturi Haldar

Editor-in-Chief

PLOS Pathogens

orcid.org/0000-0001-5065-158X

Michael Malim

Editor-in-Chief

PLOS Pathogens

orcid.org/0000-0002-7699-2064

Reviewer's Responses to Questions

**Part I - Summary**

Reviewer #1: This manuscript (no. D-23-01799) is quite interesting and novel in the way the system was examined. However, there are a number of corrections that need to be made, including some incorrect statements. These are listed further below with numerals. In addition, I think the authors need to consider more of the vast literature on this issue. Several points to consider are as follows:

A. The model look very much like the BMV replication model, where the 1a protein of BMV is produced in larger amounts than the 2a protein. That is not the case with Fny-CMV. There is always more 2a than 1a at steady-state level. In fact, in protoplast infected by Fny-CMV, the 1a protein is not even detectable by immune-blotting, although it can be detected in plant extracts. [See Virology (1995) 208: 58-66.] Thus, it is unlikely that there is excess 1a protein around to line the invaginated tonoplast membrane pocket depicted in Fig. 9. To this point, see also Discussion in Virology (2009) 383: 248-260. [Also, BMV replicates on the ER, a different membrane from CMV.]

B. In early work in the late 70’s in the lab of J.M. Kaper, it was shown that when a satRNA was replicated in tobacco, the satRNA accumulated to such a high level (as did its dsRNA) that the level of helper virus was greatly suppressed. This did not happen in squash, where there was little suppression of helper virus with moderate levels of satRNA accumulation and no high levels of satellite dsRNA. Thus, the host can influence the replication levels as much as anything else! In fact, those workers often found satRNAs in field isolates, but only after passaging them in tobacco! Thus, very low levels of satellite can be amplified under specific circumstances.

C. Also pertinent to the work here is a study from the Bruening lab, in which they found that trace amounts of TRSV-satRNA could be stimulated to replicate in the presence of non-accumulating satRNA mutants. [Virology (1995) 209: 470-479.]

D. SatRNAs also can be amplified by CMV and persist in a fungus and in an oomycete! [Eur. J. Plant Pathol. 2019, 153: 1001-1017.] This indicates that the host requirements for basic replication are minimal and likely ubiquitous.

E. It may be that the LS-CMV replicase is not as efficient as the Fny-CMV replicase. Alternatively, it may be that the LS-CMV 1a and 2a proteins are not as good at suppressing hosts defenses against the accumulation of the virus as are the Fny-CMV 1a and 2a proteins. In the former case, this then comes down to an issue of Km of the replicase for interaction with the satRNA. However, the agroinfiltrated satRNAs are not present at 0.1 fg / microgram of (virion) RNA, as were the endogenous TRSV-satRNA. In addition, while the mixed strain replicases might be less efficient than the parental virus replicases, since the 1a and 2a proteins from different strains are not adapted to each other, this is actually not the case here. Rather, both the 1a and the 2a of Fny-CMV contribute separately to the efficiency of RNA3 replication and of satRNA replication, with the LS-CMV 1a and 2a together showing the least efficiency (or most turnover).

In any event, the authors should consider these points to decide what the most likely scenario is.

Reviewer #2: The ms focuses on the role of the helper virus in the replication of satellite RNA using two strains of cucumber mosaic virus (fny and LS) that show different ability to support replication of its associated satellite Sat-1 RNA as model system. The authors have established a nice in trans replication assays to dissect the different components of the help virus that is needed for sat-1 replication. Using this system, they initially established that the fny1 replication proteins and not those of LS can support sat-1 replication and that the addition of the viral RNAs is sufficient to stimulate sat-1 replication. The topic holds significant interest as it advances our understanding on RNA sequences and protein dependence between the satellite RNA and its helper virus. The ms is well written, the experiments are nicely laid out. However, some additional controls are needed to provide support of their conclusions.

Reviewer #3: The primary objective of this study is to explore the non-templated functions of genomic RNA derived from the Cucumber mosaic virus (CMV) in the amplification of satellite RNA (sat-RNA). To achieve this objective, the researchers utilized two distinct CMV strains, CMV-Fny and CMV-LS, belonging to subgroup I and II, respectively. Additionally, the study assessed the replication efficiency of various sat-RNAs in the presence of CMV strains Fny and LS, as well as mutant sequences. The study's findings led to the conclusion that non-templated functions of CMV RNAs play a crucial role in sat-RNA replication. Unfortunately, as elaborated below, this reviewer disagrees with the conclusions, as the authors have omitted critical controls and key experiments necessary to substantiate their claims.

**Part II – Major Issues: Key Experiments Required for Acceptance**

Reviewer #1: N/A

Reviewer #2: Main comments:

The authors revealed that the non-coding version of the LS genomic RNAs are sufficient to support sat-1 replication in their trans- replication assay. However, they missed to establish sequence specificity. Could an unrelated non-coding viral sequence or non-coding Fny viral RNAs work as well in complementation with the LS replication proteins? How to explain the inability of ncL3 to support D4 replication?

The authors concluded that the LS genomic RNAs must be a replicative RNA to support sat-1 replication. However, their approaches are not “direct” and the provided data are not strong enough to fully rule out that a non-replicative RNA could still support sat-1 replication. In fact, the deletion or mutation of the TSL, 5’UTR or the box-B could result in RNA instability more than a direct interference with the RNA replication. The authors should find other ways to rule out that a non-replicative RNA is indeed non-functional for sat-1 replication: how about adding just “TLS” in trans? could the TLS alone be sufficient to support sat-1 replication? How about using sgRNA 4 as viral RNA (its non-coding version) since sgRNA 4 is non-replicative and it shares the same 3’ UTR as the gRNAs? Another option is to repeat the assay in an in-vitro replication system (as referred by the authors in their introduction). In such system, the TLS mutant and deletion mutant RNAs etc could be stable enough to measure sat-1 replication in a short time frame.

In Fig 7, the authors concluded that Ls1 is partially responsible for the defect of LS replication protein in supporting sat-1 replication. However could it be that LS1a did not support sat-1 replication because it is not compatible to form a complex with F2a? The authors should complement their experiment by looking at the ability of Ls1a+F2a (and F1a+Ls2a) to support sat-1 replication in the presence of the viral RNAs.

Line 430-460: the authors should include some binding assays or co-localization assay to support their conclusion of inability of the LS replicate to recruit satRNAs.

Reviewer #3: Before embarking on the planned experiments aimed at addressing the research questions, the authors should lay the groundwork by examining the accumulation levels of sat-RNA in the presence and absence of each HV strain. Additionally, it is imperative to consistently evaluate the levels of both plus and minus strands of sat-RNA accumulation in every experiment.

#1. Four decades ago, it was established that in the absence of its helper virus (HV), satellite RNAs (sat-RNAs) can endure for up to 25 days, as documented in Virology (94: 243-253) and Virology (86: 562-566). The molecular basis underlying the HV-independent long-term survival of sat-RNA was attributed to its ability to localize within the nucleus and exploit host enzyme activities, akin to viroids and Hepatitis delta virus (J Virology 86: 4823-32). In light of these findings, it is crucial for any study involving sat-RNA replication to incorporate a control group that solely inoculates sat-RNA and assesses the accumulation of both plus and minus strands. Notably, this vital control is conspicuously absent in all experiments conducted in this study.

#2. The data displayed in Figure 7A should be integrated into Figure 1 to assess the relative replication competence of RNA3 in the presence of heterologous RNAs 1 and 2, specifically, F1a+F2a+L3 and L1a+L2a+F3. Additionally, the authors should provide an explanation for why L3 exhibits more efficient replication with F1a+F2a compared to its homologous counterpart, L1a+L2a. Compare RNA3 and 4 levels Fig. 1A and 7A.

#3. The data presented in Figure 2 (B, D) unmistakably demonstrates that the presence of CP (comparing L3 in Figure 2B to lane ncl3 in Figure 2D) plays a substantial role in boosting sat-RNA replication. In positive-strand RNA viruses, CP has been well-documented as a regulator of strand asymmetry (as discussed in Ann Rev Phytopath 43:39-62 and Phytopathology 110: 228-236). Consequently, to verify this possibility, it is imperative to conduct an experiment that involves co-expressing CP with the inoculum as depicted in Figure 2D, lane ncL3.

#4. The data presented in Figure 3B appears to be puzzling. It's unclear why all lanes (excluding the vector lane) that do not contain satRNA are showing the presence of satRNA.

#5. In this study, the authors employed DIG-labeled DNA probes. While DIG-labeled probes offer safety and convenience, they may not be suitable for longer exposures of the blot. It is advisable to opt for radio-labeled RNA probes when extended exposures are necessary to detect poorly accumulated satRNAs and require high-stringency post-hybridization washing.

• Other major issues:

1. Throughout the manuscript, the authors consistently use the term "infection" (e.g., in lane 264, 267). It's important to note that replication and infection represent two distinct phases in the life cycle of an RNA virus. In contrast to replication, infection is a multifaceted process encompassing replication, movement, and symptom development. In this study, the authors exclusively assessed replication and did not examine the broader concept of infection. It is essential to rectify this terminology throughout the manuscript to accurately reflect the scope of the research.

2. The accumulation levels of satRNA-T1 in the presence of F1a+F2a varied significantly between experiments. Compare Fig. 1C3B. Why?

3. Unless the outcomes of the experiments recommended above are taken into consideration, this reviewer contends that the data presented in Figures 4-8 lack relevance.

4. This reviewer encountered difficulty in comprehending the model depicted in Figure 9. For instance, positive-strand RNA viruses, such as CMV, typically replicate within vesicles or spherules induced by the replicase protein of the helper virus (HV) (as discussed in Virology 383: 248-260). In the top panel of Figure 9, it remains unclear how VRC (Virus Replication Complex) formation can occur in the absence of HV RNAs. This aspect requires clarification.

5. Discussion: This section needs to be rewritten. Many of the conclusions drawn in this study rely on the results of experiments conducted in the absence of viral RNAs. However, as previously mentioned, the authors failed to evaluate the accumulation of both plus- and minus-strands of satRNA in the absence of the HV strain. Without this essential information, it becomes challenging to critically assess the non-templated functions of HV RNAs.

**Part III – Minor Issues: Editorial and Data Presentation Modifications**

Reviewer #1: Other specific issues:

1. ln 88-98. Delete “on the other hand”, since it is the same in this respect to RNA2, both of which have two ORFs.

2. ln 175 and 197. Italicize “Agrobacterium”.

3. ln 195. Change agrobacterium to Agrobacterium.

4. ln 312. Change

---

## [Decision Letter · Decision Letter 1]

17 Mar 2024

Dear Dr. Du,

Thank you very much for submitting your manuscript "The non-template functions of helper virus RNAs create optimal replication conditions to enhance the proliferation of satellite RNAs" for consideration at PLOS Pathogens. As with all papers reviewed by the journal, your manuscript was reviewed by members of the editorial board and by several independent reviewers. The reviewers appreciated the attention to an important topic. Based on the reviews, we are likely to accept this manuscript for publication, providing that you modify the manuscript according to the review recommendations.

Sincerely,

Ying Wang, Ph.D.

Academic Editor

PLOS Pathogens

Shou-Wei Ding

Section Editor

PLOS Pathogens

Michael Malim

Editor-in-Chief

PLOS Pathogens

orcid.org/0000-0002-7699-2064

Reviewer Comments (if any, and for reference):

Reviewer's Responses to Questions

**Part I - Summary**

Reviewer #2: The manuscript has been well revised. The added controls/data provided stronger support of their conclusions in the role of the CMV replicase, viral RNA motif in sat-T1 replication. The manuscript will benefit from the polishing of the english as some of sentences are too convoluted or unclear.

Reviewer #4: This manuscript by Qiao et al. presents an intriguing study on the role of non-template functions of helper virus RNAs in the replication of satellite RNAs (satRNAs), utilizing cucumber mosaic virus (CMV) and its associated satRNA as a model system. The authors provide compelling evidence that CMV RNAs can enhance satRNA replication independently of their role as templates for translation. Based on their responses to the reviewers and the revised manuscript, the authors have thoroughly addressed the major concerns and suggestions raised during the review process. They have made significant efforts to correct inaccuracies, clarify methodologies, expand on their discussions, and incorporate additional controls and experiments as requested. By adjusting the minor issues, such as term usage and grammar, this revised manuscript should be considered for acceptance.

**Part II – Major Issues: Key Experiments Required for Acceptance**

Reviewer #2: (No Response)

Reviewer #4: (No Response)

**Part III – Minor Issues: Editorial and Data Presentation Modifications**

Reviewer #2: I will encourage the authors to polish their english. Some sentences such as line 290-291 are unclear.

The conclusion is line 384 needs to be edited. Isn’t the accumulation of negative strand stat 1 (-) RNA the evidence for sat-1 replication? If so, the conclusion seems incorrect. Couldthe accumulation of sat1(+) could be increase of stability of the RNA.

Reviewer #4: (No Response)

PLOS authors have the option to publish the peer review history of their article (what does this mean?). If published, this will include your full peer review and any attached files.

Reviewer #2: No

Reviewer #4: No

Figure Files:

Data Requirements:

Reproducibility:

References:

---

## [Editor Report · Decision Letter 2]

7 Apr 2024

Dear Dr. Du,

We are pleased to inform you that your manuscript 'The non-template functions of helper virus RNAs create optimal replication conditions to enhance the proliferation of satellite RNAs' has been provisionally accepted for publication in PLOS Pathogens.

Best regards,

Ying Wang, Ph.D.

Academic Editor

PLOS Pathogens

Shou-Wei Ding

Section Editor

PLOS Pathogens

Michael Malim

Editor-in-Chief

PLOS Pathogens

orcid.org/0000-0002-7699-2064
---

## [Editor Report · Acceptance letter]

12 Apr 2024

Dear Dr. Du,

We are delighted to inform you that your manuscript, "The non-template functions of helper virus RNAs create optimal replication conditions to enhance the proliferation of satellite RNAs," has been formally accepted for publication in PLOS Pathogens.

Best regards,

Michael Malim

Editor-in-Chief

PLOS Pathogens

orcid.org/0000-0002-7699-2064